

**Hydrochemical and isotopic evidences for deciphering conceptual model of groundwater**
**salinization processes in a coastal plain, north China**
Han Dongmei[a,b]
a Key Laboratory of Water Cycle & Related Land Surface Processes, Institute of Geographic Sciences and Natural Resources
Research, Chinese Academy of Sciences, Beijing, 100101, China
b College of Resources and Environment, University of Chinese Academy of Sciences, Beijing 100049, China
**Abstract**
Groundwater is the important water resource for agricultural irrigation, urban and tourism development and
industrial utilization in the coastal region of north China. In the past five decades, coastal groundwater
salinization in the Yang-Dai River coastal plain has become more serious than ever before under natural
climate change and anthropogenic activities. It is pivotal for the scientific management of coastal water
resources to accurately understand groundwater salinization processes and its inducement. Hydrochemical
and stable isotopic ($\delta^{18}O$ and $\delta^2H$) analysis for the different water bodies (surface water, groundwater,
geothermal water, and seawater) were applied to provide a better understanding of the processes of
groundwater salinization in the Quaternary aquifers. Saltwater intrusion is the major aspect and can be
caused by vertical infiltration along the riverbed at the downstream areas of rivers during the tide/surge
period, and lateral inflow into fresh aquifer derived from intensively pumping groundwater. Seawater
proportion can reach ~13% in the well field. High mineralized geothermal water (TDS up to 10.6 g/L) with
the indicator of paleoseawater relics (lower Cl/Br ratios relative to modern seawater) overflows into the
cold Quaternary aquifers. Groundwater salinization can also be exacerbated by the anthropogenic activities
(e.g., irrigation return-flow with solution of fertilizers, domestic wastewater discharge). Additionally, the
interaction between surface water and groundwater can make the groundwater freshening or salinizing in
different sections to locally modify the groundwater hydrochemistry. The cease of the well field and
establishment of anti-tide dam in the Yang River estuary area have effective function to contain the
development of saltwater intrusion. This study can guide the future water management practices, and
provide research approaches and foundation for further investigation of seawater intrusion in this and
similar region.
**1. Introduction**

Coastal region is the key area for the world social and economic development. Approximately 40% of

the world's population lives within 100 kilometers of the coast (UN Atlas, 2010). The worldwide coastal



area has become increasingly urbanized with 14 of the world's 17 largest cities located along coasts (Creel
L., 2003). China has 18,000 km of continental coastline, about 164 million people (about 12% of total
Chinese population) live in 14 coastal provinces, and nearly 80% of them distribute in the three coastal
economic regions, namely Beijing-Tianjin-Hebei economic region, the Yangtze River delta economic
region and the Pearl River delta economic region (Shi, 2012). The rapid economic development and the
growing population in the coastal region have increased demands for fresh water, meanwhile been
confronted with the threat from waste and sewage water discharge into coastal ecosystems.

Coastal groundwater resources play crucial roles on the social economic and ecologic function in the

global coast system (IPCC, 2007). Coastal groundwater system connects the ocean and the continental
hydro-ecological systems (Moore, 1996; Ferguson and Gleeson, 2012). Groundwater as an important
freshwater resource could be over extracted due to that the periods of highest demand (e.g., agricultural
irrigation and tourist seasons) are often the periods of lowest recharge rates (Post, 2005). In addition to
occurrence of some environmental issues, such as land subsidence, contaminants transport, the
over-exploitation of groundwater can readily result in seawater intrusion in the coastal area. Seawater
intrusion has become a global issue and the related studies can be found from the coastal aquifer system of
different countries, such as Israel (Sivan et al., 2005; Yechieli et al., 2009; Mazi et al., 2014), Spain (Price
and Herman, 1991; Pulido-Leboeuf, 2004; Garing et al., 2013), France (Barbecot et al., 2000; de Montety
et al., 2008), Italy (Giambastiani et al.., 2007; Ghiglieri et al., 2012), Morocco (Bouchaou et al., 2008; El
Yaouti et al., 2009), USA (Gingerich and Voss, 2002; Masterson, 2004; Langevin et al., 2010), Australia
(Zhang et al., 2004; Narayan et al., 2007; Werner, 2010), China (Xue et al.., 2000; Han et al., 2011, 2015),
Vietnam (An et al., 2014), Indonesia (Rahmawati et al., 2013), India (Radhakrishna, 2001; Bobba, 2002),
Brazil (Gico Montenegro et al., 2006; Cary et al., 2015), etc. Werner et al. (2013) gave an excellent review
on seawater intrusion processes, investigation and management. A variety of approaches have been used to
investigate seawater intrusion, including head measurement, geophysical methods, geochemical methods
(environmental tracers combined hydrochemical and isotope data), conceptual and mathematical modeling
(see reviews by Jones et al., 1999; Werner et al., 2013).

Seawater/saltwater intrusion is a complicated hydrogeological process due to the impact of aquifer

properties, anthropogenic activities (e.g., intensive groundwater pumping, irrigation practices), recharge
rate, variable density flow between the estuary and adjacent fresh groundwater system, tidal/surge activity
and global climate change (Ghassemi et al., 1993; Robinson et al., 1998; Smith and Turner, 2001; Simpson





and Clement, 2004; Narayan et al., 2007; Werner and Simmons, 2009; Wang et al., 2015). Brockway et al.
(2006) reported the negative relationship between saltwater intrusion length and river discharge.
Understanding the complex interactions between groundwater and surface water, groundwater and seawater
is essential for the effective management of water resources (Sophocleus, 2002; Mondal et al., 2010). There
was a vastly different result based on numerical simulations for the additional distance of intrusion in the
Nile Delta Aquifer of Egypt and in the Bay of Bengal under the same sea level rise (Sherif and Singh,
1999). Bobba (2002) also employed numerical simulations to demonstrate an apparent risk of saltwater
intrusion in the Godavari delta, India due to sea level rise. Westbrook et al. (2005) defined the hyporheic
transition zone of mixing between river water and groundwater influenced by tidal fluctuations and the
contaminant distribution. Modelling seawater intrusion in the Burdekin Delta irrigation area, North
Queensland (Australia) show that seawater intrusion is far more sensitive to pumping rates and recharge
than to aquifer properties (e.g., hydraulic conductivity), and compared to the effects of groundwater
pumping, the effect of tidal fluctuations on saltwater intrusion can be neglected (Narayan et al., 2007).
However, rare studies have focused on delineating the interactions among surface-ground-sea-waters in
estuarine environment and the effects of vertical infiltration of seawater into the off-shore aquifer through
river channel vs. the lateral landward migration of the freshwater-saltwater interface.

The data of China's marine environment bulletin released on March 2015 by State Oceanic

Administration People's Republic of China showed that the major bays, including Bohai Bay, Liaodong
Bay, Hangzhou Bay, are polluted seriously with the inorganic nitrogen and active phosphate as the major
pollutants (SOA, 2015). Seawater intrusion in China is the most serious around the Circum-Bohai-Sea
region. The escalating seawater intrusion in the future may be not a simple problem related to groundwater
salinization in this region. It is likely to be more difficult to remediate groundwater pollution caused by the
contaminated seawater. This study will take the Yang-Dai River coastal plain in Qinhuangdao City, Hebei
province of north China, as an example to investigate groundwater salinization processes and interactions
among surface water, groundwater and seawater, and the seawater intrusion caused by groundwater
exploitation in Zaoyuan well field. Qinhuangdao is an important port and tourist city of northern China. In
the past 30 years, many previous studies had done to investigate distribution of seawater intrusion and its
influence factors using hydrochemical analysis of groundwater (Xu, 1986; Yang, 1994, 2008; Chen and Ma,
2002; Sun and Yang, 2007; Zhang, 2012) and numerical simulations (Han, 1990; Bao, 2005; Zuo, 2009).
This study is a continuation of previous investigations of the coastal plain aquifers in Qinhuangdao.



Hydrochemical and stable isotopic compositions of collected water samples were analyzed for making up
the knowledge gap of surface-, ground-and sea-water interactions in this region. This study aims to describe
the conceptual model of the complex processes for the groundwater salinization of the coastal aquifers, to
reveal the major aspects responsible for the increasing groundwater salinity in the coastal aquifers, and to
obtain a conceptual model for deciphering the groundwater flow system of the study area. The results will
be helpful for the further numerical simulations of coastal groundwater system. It is very significant for
water resources management in the coastal plain.
**2. Study area**
The Yang-Dai River coastal plain (Fig. 1) covers approximately 200km$^2$ in the west side of Beidaihe
District of Qinhuangdao City, the northeastern Hebei Province. It connects the eastern section of Yanshan
Mountain and surrounded by mountains. The southern boundary of the study area is Bohai Sea. The plain
become low from northwest to southeast and fan-shaped distribution of the piedmont- coastal inclined
alluvial plain. Elevation ranges from 390 in the west and to 40-100 m in the north, and 25-40 in the east,
and 25-1m in the south coastal region, with the average slope of 0.008. Zaoyuan well field, located in the
southern edge of alluvial fan, was built in 1959 (Xu, 1986) as major water supply for this region. It is 4.3
km from the southeastern well field to Yang River estuary.
**2.1 Climate and hydrology**
The study area is in a warm and semi-humid monsoon climate. On the basis of a 56-a record in
Qinhuangdao area, the mean annual rainfall is estimated to be 640 mm, the average annual temperature is
about 11°C, and mean potential evaporation of 1469 mm. 75% of the total annual rainfall falls in
July-September (Zuo, 2006). The average annual tide level is 0.86m (meters above Yellow Sea base level),
the highest tide is 2.48m, and the lowest is -1.43m.
The Yanghe River and Daihe River originated from the Yanshan Mountains are the major surface
water body in this area (Fig. 1). The river soared when heavy rains happened with short peak duration,
whereas it became minimal flow or drying during the dry season. The Yang River is about 100 km long
with the catchment area of 1029 km$^2$, and the average annual runoff of $1.11\times10^8$ m$^3$/a (Han, 1988). Dai
River has the length 35 km and catchment area of 290 km$^2$, with annual runoff of $0.27\times10^8$ m$^3$/a and
average gradient of 11.4‰. The two rivers flow into the southern Bohai Sea.



## 2.2 Geological and hydrogeological setting

Groundwater in this area mainly include fissure water in the bedrock and water in the Quaternary porous media. The bedrock fissure water is distributed in the north platform area. Its water abundance is mainly depended on the degree of weathering and the nature and regularity of fault zone. The strata outcropping in the west, north and eastern edge of the plain includes the Archean gneiss, Proterozoic mixed granite and Jurassic sandstone, shale and so on. The ex-Quaternary, which is exposed in the offshore area of the region, is mainly the Archean metamorphic granite, which is widely distributed. The mineral composition includes mainly quartz, feldspar, and biotite. The Quaternary sediments of the plain are mostly underlain by the Archean gneisses and Proterozoic mixed granites. The basement faults under the Quaternary cover mainly include the NE-trending fault and the NW-trending fault. The Quaternary aquifer system of the Yang-Dai River coastal plain is a complete groundwater system from the piedmont to the coast (see P-P' cross-section of Figure 2). Geological technics control the development and deformation of the sediments, the distribution of hot springs and geothermal anomalies. Fault zones are also the main channel for deep-water cycle and thermal convection.

The Quaternary sediments are widely distributed in the area. The bottom of the Holocene in most areas has clay or clay layers, which make the groundwater in the coastal zone under confined or semi-confined status. There are no regional, continuous aquitards between several aquifers. The thickness of the Quaternary strata has a range of 5-80 m, mostly 20-40 m, and up to more than 100 m near the coastline. The aquifer is mainly composed of medium sand, coarse sand and gravel with thickness of 10-20 m and water table depth of 1-4 m in the phreatic aquifer, and thickness of 10-30 m and water table depth of 1-5 m in the confined aquifer (Zuo, 2006). In the yearly peak season of agricultural water, the groundwater level decline sharply and reaches the lowest water table in April-May period, and become highest in January-February. The main sources of aquifer recharge are from rainfall infiltration, river water and irrigation return-flow, lateral subsurface runoff from the piedmont area. Apart from the phreatic water evaporation, groundwater pumping it the main pathway of groundwater discharge for agricultural, industrial, tourism and sanatorium's utilization. The general flow direction of groundwater is from northwest to south. Naturally, groundwater discharges into the river and the Bohai Sea.

The geothermal water near the fault zone discharges into shallow Quaternary sediments, which is the overlying strata in geothermal anomalous area (Hui, 2009). The temperature of thermal water range is 27-57 ℃ in this low-to-medium temperature geothermal field (Zeng, 1991). The thickness of the overlying



strata is varied from 24.6 to 58.8 m and consists of alluvial sand, gravel, clayey loam, clay and silt. The
thermal water stored in the Archaeozoic granite and metamorphic rocks, which are composed of migmatite,
gneiss, and amphibole plagio-gneiss (Pan, 1990). Major deep fracture zones are the good passage for the
geothermal water movement (Yang, 2011). The heated groundwater in the deep zones could upward
transport along the fault and mix with the cold groundwater in the Quaternary aquifers (Shen et al., 1993).
**2.3 Environmental issues and seawater intrusion history**
The shallow groundwater pumped from the Quaternary aquifer occupies 94% of the total groundwater
exploitation, which is used for agricultural irrigation (accounts for 52% of the total groundwater use),
industrial (32%) and domestic water (16%) (Meng, 2004). Many large and medium-sized reservoirs were
built in the 1960s and 1970s and resulted in that the surface water was intercepted and the downstream
runoff dropped sharply, even became dry in drought years. With the intensification of human
socio-economic activities and growing urbanization, coupled with extended drought years (severe drought
during 1976-1989 in north China) (Wilhite,1993; Han et al., 2015), increased groundwater exploitation to
meet the ever-growing fresh water demands has resulted in groundwater level declining and seawater
intrusion (SWI) in the coastal aquifers.
The pumping rate in the Zaoyuan well field was gradually increased from 1.25 million $m^3$/a in the
early 1960s to 3.5 million $m^3$/a in the late 1970s, and beyond 10 million $m^3$/a in the 1980s. During
1966-1989, the major agricultural planting in this region is paddy field with big water consumption. The
groundwater pumping time is mainly from May to October with pumping rate of 7~80,000 $m^3$/d, which
was over-exploited and resulted in formation of groundwater level declining depression. In May 1986, the
groundwater level in the depression center, which is located in Zaoyuan-Jiangying, was decreased to below
-2 m.a.s.l. (meters above sea level), with the depression area, which has groundwater level below the sea
level, covered 28.2 $km^2$. Since 1990, the rapid development of township enterprises in the 1980s (mainly
refer to paper mills), groundwater over-exploitation in the west area (i.e. the groundwater pumping rate for
paper mill development reached 55,000 $m^3$/d in 2002) resulted in the groundwater level depressions around
Liushouying and Fangezhuang. The lowest groundwater level in the depression center in 1991 was up to
-11.6 m.a.s.l., and -17.4 m.a.s.l. in 2002. After the implementation of "Transfering Qing River water to
Qinhuangdao" project since 1992, the intensity of groundwater pumping became slowed down. The
depression center moved to Liushouying area. The groundwater level of the depression center was





recovered to -4.3 m.a.s.l. in July 2006. The shape of the depression was elliptical with the major axis of the
EW direction. The depression area developed to 132.3km$^2$ in May 2004.

The groundwater quality of this area has become gradually salinized since the early 1980s. Chloride

concentrations increased year by year. As early as 1979, seawater intrusion occurred in the Zaoyuan well
field. The intrusion area with groundwater chlorine concentration greater than 250 mg/L has been
developed to 21.8 km$^2$ in 1984, and 32.4 km$^2$ in 1991, 52.6 km$^2$ in 2004, 57.3 km$^2$ in 2007 (Zuo, 2006;
Zang et al., 2010). The chloride concentration of groundwater pumped from this well field changed from 90
mg/L in 1963, 218 mg/L in 1978, 567 mg/L in 1986, 459 mg/L in 1995, and 1367 mg/L in 2002 (Zuo,
2006), 812 mg/L in July 2007 (this study). The distance of seawater intrusion into the inland reached 6.5
km inland in 1991, and developed to 8.75 km in 2008 (Zang et al., 2010). At the early 1990s, 16 of 21
pumping wells in the well field have been abundant due to the salinized water quality (Liang et al., 2010).
370 of 520 pumping wells has been abundant in the Yang-Dai River coastal plain during 1982-1991 (Zuo,

2006).

**3. Methods**

Totally 80 water samples were collected from the Yang-Dai River coastal plain, including 58

groundwater samples, 19 river water samples (from 12 sites), and 3 seawater samples during three sampling
campaigns, namely, June 2008, September 2009 and August 2010. Groundwater samples pumped from 28
productive wells with well depth of 6-110m, including 7 deep wells, which has well depth more than 60m.
The water sampling sites can be shown in Figure.1. In this study, we investigated cold groundwater from
the productive wells. However, the geothermal water existing around Danihe cannot be ignored. The related
data can be available and referenced from Zeng (1991) due to that we cannot obtain the hot water samples
from the current geothermal field.

Measurements of some physical-chemical parameters (i.e. pH, temperature, and electrical conductivity

(EC)) were conducted in situ using portable meter (WTW Multi 3500i). All water samples were filtered to
0.45 μm membrane filters before collection for analysis of hydrochemical composition. Two aliquots in
polyethylene 100mL bottles at each site were collected: for major cation and anion analysis, respectively.
Samples for cation analysis (Na, K, Mg and Ca) were added 6 N HNO$_3$ to prevent precipitation. Water
samples were sealed and stored at 4 ºC until determination. Biocarbonates were determined by titration
within 12 hours after sampling. The concentrations of cations and some trace elements (i.e. B and Sr) were



analyzed by inductively coupled plasma-optical emission spectrometry (ICP-OES) on filtered samples in
the chemical laboratory of the Institute of Geographic Sciences and Natural Resources Research (IGSNRR),
Chinese Academy Sciences (CAS). Concentrations of major anions (i.e. Cl, $SO_4$, $NO_3$ and F) were analyzed
by a High Performance Ion Chromatograph (SHIMADZU, LC-10ADvp) at the IGSNRR, CAS. The ion
balance errors of the chemical results are less than 8%. The hydrochemical and physical data are shown in
Table 1. The stable isotopes ($\delta^{18}O$ and $\delta^2H$) of water samples were measured by a Finnigan MAT 253 mass
spectrometer after on-line pyrolysis with a Thermo Finnigan TC/EA in the Stable Isotopes Laboratory of
the IGSNRR, CAS. The results of $\delta^{18}O$ and $\delta^2H$ values shown in Table 1 were expressed in ‰ relative to
international standards (V-SMOW (Vienna Standard Mean Ocean Water)). The analytical precision for $\delta^2H$
is ±2‰ and for $\delta^{18}O$ is ±0.5‰.

Saturation indices for common minerals (i.e. calcite, dolomite, and gypsum) were calculated using

PHREEQC version 2.8 (Parkhurst and Appelo, 1999) to understand the saturation status of these minerals
in the aquifer. Ionic delta values were calculated to further investigate the hydrogeochemical behavior that
take place in the aquifer and modify groundwater hydrochemistry. The ionic delta values express
enrichment or depletion of each ion's concentration relative to its theoretical concentration calculated from
the Cl⁻ concentration of the sample for a conservative freshwater-seawater mixing system (Fidelibus et al.,
1993; Appelo, 1994). The delta values have been used as effective indicators of coastal groundwater
undergoing freshening or salinizing processes, accompanied by related water-rock interaction (prevailingly
cation exchange). Cl⁻ can be regarded as a conservative tracer for the calculations mentioned below. The
seawater contribution for each sample can be expressed by a fraction of seawater ($f_{sw}$), which can be
calculated using (Appelo and Postma, 2005):

$$f_{sw} = \frac{C_{Cl,sam} - C_{Cl,f}}{C_{Cl,sw} - C_{Cl,f}} \qquad (1)$$

where $C_{Cl,sam}$, $C_{Cl,f}$, and $C_{Cl,sw}$ refer to the Cl⁻ concentration in the sample, freshwater, and seawater,

respectively. Based on the $f_{sw}$ value, the theoretical concentration ($C_{i,mix}$) of each ion in a water sample can
be calculated by:

$$C_{i,mix} = f_{sw} \cdot C_{i,sw} + (1 - f_{sw}) \cdot C_{i,f} \qquad (2)$$

$C_{i,sw}$, and $C_{i,f}$ refer to the measured concentration of the ion $i$ in the seawater and freshwater,

respectively. The ionic delta value ($\Delta C_i$) of ion $i$ can be obtained by:





$$\Delta C_i = C_{i,sam} - C_{i,mix} \qquad (3)$$


$C_{i,sam}$ - the measured concentration of the ion $i$ in the water sample.
**4. Results**
4.1 Groundwater dynamics
Due to the different groundwater pumping rate and patterns, the variation trend of groundwater level
has been different in the east and west areas of the Yang-Dai River coastal plain. In the east part, owing to
the intensive exploitation in the Zaoyuan well field, the groundwater level was gradually declined to be
lower than the sea level during the 1980s. The center of groundwater level depression was located in
Zaoyuan-Jiangying region, with the groundwater level lower than -3 m.a.s.l. The local government
commenced to reduce the exploitation after 1992. The groundwater level decreased slowly after 1995, even
started to recovery in some wells as a result of pumping reduction. During the extreme drought year (1999),
the consequential increased water demand made the groundwater level declined again in the east region. In
the late 1980s, the groundwater level at the west region was still more than 0 m.a.s.l. But in the late 1990s,
due to the fast development of the local paper mills as the big water consumers, the groundwater level
dropped year by year and had big falling amplitude after 2000, resulting in the overall transfer of
groundwater depression center to the western region (Liushouying-Fangezhuang). The groundwater level in
this center was up to -14 m.a.s.l. in May 2002.
Based on the data from the three monitoring wells, the seasonal variation of groundwater level in this
area can be seen from Figure 3. After 2000, the groundwater level in the east of the Yang-Dai River coastal
plain was mainly affected by the groundwater pumping for agricultural and domestic water use. During
March and June of each year, the shallow groundwater pumping as the major water source for irrigation has
resulted in the fast dropped water level occurred between April and June, down to the lowest level of water
throughout the year. As the rainy season started in July, groundwater pumping began to decrease.
Groundwater level rise rapidly with the infiltration of irrigation return-flow and rainfall, lateral subsurface
runoff from the surrounding aquifers. After the end of the rainy season (July to September), the water level
continues to rise gently and reach the annual maximum water level during January and February. With the
amount of recharge is reduced along with the increase of domestic water pumping, water level circularly
slow down to the next agricultural peak. In addition to groundwater over-exploitation, climate
change-induced recharge reduction in recent three decades has been also part of the cause of groundwater





level declining, resulting in the seawater intrusion. The annual average rainfall varied from 639.7 mm
(1954 - 1979) to 594.2 mm (1980-2010). It obviously finds that there is a significant decrease in rainfall
over the last 30 years (Zhang, 2012). In general, the groundwater runoff intensity gradually decreases from
the piedmont to the coastal region.
4.2 Water stable isotopes ($\delta^2$H and $\delta^{18}$O)

19 water samples collected from Yang River and Dai River have $\delta^{18}$O and $\delta^2$H values ranging from

-10.1 to -0.6‰ (mean= -5.4‰) and from -71~-11‰ (mean = -43‰), respectively. It seems that the stable
isotopes composition have significant seasonal variation. For Yang River, 3 surface water sampled in
relatively dry season (June 2008) were characterized by $\delta^{18}$O and $\delta^2$H values ranging from -5.5 to -1.1‰
(mean= -3.0‰) and from -49~-15‰ (mean = -31‰), respectively. Whereas 6 water samples sampled in
wet season (August 2009 and September 2010) have $\delta^{18}$O and $\delta^2$H values ranging from -10.1 to -2.4‰
(mean= -6.6‰) and from -71~-21‰ (mean = -48‰), respectively. As to Dai River, in dry season, 3 surface
water samples are characterized by $\delta^{18}$O and $\delta^2$H values ranging from -3.9 to -0.6‰ (mean= -2.6‰) and
from -44~-11‰ (mean = -32‰), respectively; and in wet season, 7 surface water samples have $\delta^{18}$O and
$\delta^2$H values ranging from -9.7 to -1.2‰ (mean= -6.6‰) and from -69~-12‰ (mean = -49‰), respectively.
The water samples collected from Yang River and Dai River have similar stable isotopes composition.

56 groundwater samples are characterized by $\delta^{18}$O and $\delta^2$H values ranging from -11.0 to -4.2‰

(mean= -6.5‰) and from -76~-39‰ (mean = -50‰), respectively. Among them, 43 shallow groundwater
samples have $\delta^{18}$O and $\delta^2$H values ranging from -11.0 to -4.2‰ (mean = -6.6‰) and from -76~-39‰
(mean = -50‰), respectively; 13 deep groundwaters have $\delta^{18}$O and $\delta^2$H values ranging from -7.8 to -5.1‰
(mean = -6.3‰) and from -58~-43‰ (mean = -50‰), respectively. For the shallow groundwater, during the
dry season, 12 water samples have $\delta^{18}$O and $\delta^2$H values ranging from -7.2 to -4.2‰ (mean = -5.7‰) and
from -56~-39‰ (mean = -48‰), respectively; during the wet season, 31 water samples are featured by
$\delta^{18}$O and $\delta^2$H values with a range of -11.0 ~ -5.3‰ (mean = -6.9‰) and -76 ~ -43‰ (mean = -51‰),
respectively. For the deep groundwater, during the dry season, 3 water samples have $\delta^{18}$O and $\delta^2$H values
ranging from -5.3 to -5.1‰ (mean = -5.2‰) and from -47~-45‰ (mean = -46‰), respectively; during the
wet season, 10 water samples are featured by $\delta^{18}$O and $\delta^2$H values with a range of -7.8 ~ -5.2‰ (mean =
-6.6‰) and -58 ~ -43‰ (mean = -51‰), respectively.

The local meteoric water line (LMWL, $\delta^2$H=6.6 $\delta^{18}$O+0.3, n=64, r$^2$=0.88) is based on $\delta^2$H and $\delta^{18}$O



mean values of the monthly rainfall between 1985 and 2003 at Tianjin station some 120 km SW of
Qinhuangdao City. The data were obtained from International Atomic Energy Agency/World
Meteorological Organization (IAEA/WMO, 2006). Due to the similar climatic and coastal conditions
between Tianjin and Qinhuangdao, this meteoric water line can be regarded as the local meteoric water line
(LMWL) in this study. From Figure 4, it can be seen that surface water have more wide range of $\delta^{18}O$ and
$\delta^2H$ values relative to groundwater. Water samples collected in wet season have more wide range of $\delta^{18}O$
and $\delta^2H$ values relative to water sampled in dry season. Most of water samples plot blow the LMWL.
Seawater represents an end-member enriched in isotopes and plots far below the LMWL.
4.3 Water salinity and major dissolved ions
TDS (total dissolved solids) concentrations of the surface water samples from Dai River have a range
of 0.3g/L~31.4g/L with 22-78% $Na^+$, 56-4% $Ca^{2+}$ of total cations and 36-91% $Cl^-$ of total anions,
Ca•Na•Mg-Cl•$HCO_3$ to Na-Cl water type from the upstream to the downstream locations. The $Cl^-$
concentrations varied from about 70 mg/L in the upstream to 16700 mg/L near the coastline. For Yang
River, the collected water samples have TDS concentrations of 0.3-26.1 g/L with percentages (33-91%) of
$Cl^-$ concentrations (63.2-14953.5 mg/L) from the up-reach to the down-reach locations, with water types
changed from Ca•Na-$HCO_3$•Cl•$SO_4$, Ca•Mg-Cl•$SO_4$•$HCO_3$ to Na-Cl. The nitrate contents range from 2.8 to
65.2 mg/L in the surface water samples.
Groundwater hydrochemistry can be modified the comprehensive effects from geological, climatic,
hydrogeological processes and anthropogenic activities. In the early 1960s, groundwater pumped from the
Zaoyuan well field was featured by the Ca-$HCO_3$ water type and chloride concentrations of 90-130 mg/L.
In the early 1970s, individual wells appear slightly salinized. It has been deteriorated rapidly since the early
1980s. The chloride concentration of groundwater from water supply wells was 90mg/L in 1963, 218 mg/L
in 1975, 385 mg/L in 1984, 456.3 mg/L in 1986, 459.5 mg/L in 1995, 928.3 mg/L in 2000, 1367 mg/L in
2002, and 1290.4 mg/L in 2005 (Zang et al., 2010). In this study, the shallow groundwater is characterized
by TDS concentrations of 0.4-4.8 g/L with the percentage of $Cl^-$ (34-77%), $Na^+$ (12-85%), $Ca^{2+}$ (5-69%)
and water types varied from Ca-$HCO_3$•Cl, Ca•Na-Cl, Na•Ca-Cl to Na-Cl, which can been seen from Piper
plot (Figure 5). The deep groundwater is featured by TDS concentrations of 0.3-2.8g/L, which is dominated
by Ca (up to 77%) in the upstream area and Na (up to 85%) near coast, with water type distributed in series
of Ca-Cl•$HCO_3$, Ca•Na-Cl and Na•Mg-Cl (Figure 5). The TDS of groundwater from the well field reaches



3.31 g/L with Na-Cl water type in the well G15. The relative high fracture of seawater occurs in the shallow
well G10 and deep well G2, with *fsw* values of 12.95% and 5.35%, respectively. The nitrate contents have a
range of 2.0-178.5 mg/L (mean 90.1 mg/L) for shallow groundwater, and 2.0-952.1 mg/L (mean 232.1
mg/L) for the deep groundwater, respectively, most of which seriously exceeds the WHO drinking water
standard (50 mg/L).
There is a geothermal field around Danihe-Luwangzhuang area (Fig. 1). Hydrochemical features of
thermal water are very distinct from cold water. The previous investigation has identified the buried
geothermal water with high TDS in the fracture/fissure of deep metamorphic rock (Zeng, 1991). Due to the
pumping wells for pumping thermal water were protected and not permitted to be sampled, we have to
collect some data associated this geothermal field from the previous research. The geothermal field is
controlled by the fault distribution under confined state. The thermal water flows along the fault zone and
enters the Quaternary aquifer, forming hot salt water distributed around the spill point and expanded
towards downstream. It can result in the similar hydrochemical characteristics between Quaternary salt
groundwater and deep original thermal waters from bedrocks. The geothermal water is characterized by
Ca•Na-Cl water type, 6.2-10.6 g/L of TDS and 7.4-8.7 of pH values. $Cl^-$ concentrations range from 5.4 to
6.5 g/L, $Na^+$ from 1.7 to 2.0 g/L, $Ca^{2+}$ from 1.6 to 1.9 g/L, $F^-$ from 3.0 to 3.6 mg/L, Sr from 6.73 to 89.8
mg/L, Li from 0.43 to 1.58 mg/L, and $SiO_2$ from 44.0 to 48.3 mg/L (Hui, 2009). The groundwater samples,
collected from the wells G8, G19, and G9 with different depths, are featured by Ca•Na-Cl water type with
relative high TDS ranges (0.8-1.4 g/L, 1.3-1.6 g/L, and1.5-2.8 g/L, respectively) and Sr contents (1.1-1.9
mg/L, 4.9-7.1 mg/L, and 7.3-11.6 mg/L, respectively).
**5. Discussions**
5.1 Groundwater flow system and hydrochemical features
Generally, Quaternary groundwater system in the Yang-Dai River coastal plain is recharged by
precipitation, irrigation return flow, river infiltration and lateral subsurface runoff from mountain-front
region. Due to the natural geological function and human pumping activities, there have been interactions
between groundwater and geothermal waters around Danihe area or between groundwater and seawater in
the coastal area. The groundwater geochemical features are controlled by the complex hydrogeological
conditions and these hydrological processes. The different sources of water bodies are characterized by
different of stable isotopic and hydrochemical compositions, determining the groundwater salinization



processes in this area.

The stable isotopes of O and H can be used to describe the groundwater origin and to identify the

mixing processes between different water bodies. The slope of the best-fit regression line for collected
groundwater samples (dashed line in Fig. 4) given as $\delta^2H=4.4\times\delta^{18}O-21.7$, which is significantly lower than
either the local or global meteoric water lines. The deviation of groundwater and surface water lines from
the LMWL has evidenced evaporative processes occurred during water infiltration and surface runoff. The
composition of stable isotopes in groundwater samples collected in the relatively dry season has been
narrower and enricher than that collected in the wet season. It could be resulted from evaporation processes
during the infiltration of local irrigation return-flows in the dry season.

The composition of stable isotopes for thermal groundwater could be originated from precipitation,

however, its $^{14}C$ age dating between 3.4-12.8ka with tritium content of less than 2 TU (Zeng, 1991),
indicating thermal waters might be formed under cooler climate condition than present climate. The
composition of stable isotopes of thermal groundwater are more depleted than that of cold groundwater,
even lower than the cold groundwater from mountain-front area, indicating the thermal groundwater could
be mainly originated from NW mountain area, where has higher elevation. The elevation range of recharge
area for Danihe geothermal field is from 1200 to 1500 m.a.s.l obtained by Zeng (1991).

Fresh groundwater has depleted $\delta^{18}O$ and $\delta^2H$ values relative to seawater. Theoretically, the mixing of

fresh groundwater and seawater should show a straight line connecting the two end members. Obviously,
some surface water samples (e.g. S1, S2, S3, S7, S12) are the mixture with seawater. In this study area,
there are three end members (namely, fresh groundwater, thermal groundwater and seawater), which has
been evidenced by the previous studies (Han, 1988; Zeng, 1991). Thus, the diagram of $\delta^{18}O$ vs. $Cl^-$ (Fig. 6)
can be used to identify the mixing pattern among three end members. Fig. 6 shows the mixing lines
between shallow fresh groundwater (G4) and seawater, between deep fresh groundwater (G25) and
seawater, between shallow fresh groundwater and thermal water, and between deep fresh groundwater and
thermal water. The shallow groundwater samples (e.g. G15, G10, G11, G14) collected from or around the
Zaoyuan well field are characterized by mixing with seawater. The deep groundwater samples (e.g. G13,
G2, G16, G14') collected from the coastal zone are also resulted from mixing with seawater. The sampling
site of deep groundwater sample G29 is located between thermal field and the coastline and obviously
affected by both of mixing processes. The groundwaters (e.g. G9, G19), sampled from the area affected by
geothermal field are mixture between fresh cold groundwater with thermal waters. The mixing fraction ($f_{sw}$)



of seawater has a range of 1.2~13.0% for the shallow brackish groundwater, and 2.6~6.0% for the deep
brackish groundwater. $f_{sw}$ reaches the highest percentage of 13% in the well G10, which is located in the
north part of the well field.

At the late 1950s, groundwater pumped from the Zaoyuan well field was characterized by Ca-HCO$_3$

type water with Cl concentrations ranging from 130 to 170 mg/L. The hydrochemical data investigated in
1986 (Han, 1986) showed that there were mainly five water types in this study area, including Ca-HCO$_3$
type with TDS less than 0.5g/L distributed in the mountain-front area, Ca•Na-Cl•SO$_4$, Ca•Na•Mg-SO$_4$•Cl,
or Na•Ca-Cl•SO$_4$ type water with TDS 0.4-0.7g/L distributed around the Zaoyuan well field and
Wanggezhuang, Ca•Na-Cl type water with TDS 0.4-1.8g/L distributed around the geothermal field
(Luwangzhuang) and Duzhai, Na-HCO$_3$ or Na-HCO$_3$•Cl type water with TDS 0.5-0.9g/L distributed the
SW area close to the coastal zone, and Cl-Na type water with TDS 0.4-2.4g/L distributed in the coastal
zone. Due to the disturbance of human activities, the current groundwater hydrochemitry has become more
complex than that before. Compared salty water distributed 2 km away from coastline in the late 1950s, the
distance has increased to about 7 km away from coastline. The Cl-Ca•Na or Cl-Ca type water type mainly
distributed in the area affected by geothermal field, such as G5, G8, G19, G29, and G24, indicating the
salinizing process during the mixing between cold groundwater and the thermal waters. In the upstream
area, the groundwater samples (e.g. G7, G23, G25) have feature of Ca•Mg•Na-Cl•SO$_4$, Ca-Cl•SO$_4$, and
Ca-Cl•HCO$_3$ type, not the Ca-HCO$_3$ type in the 1980s. It suggests that the salinized composition has
resulted from the anthropogenic pollution. The groundwater samples (e.g. G10, G11, G15, G26) collected
from the well field show the feature of Cl-Na• (Ca) type water with TDS 1.2-4.8 g/L. The samples (e.g. G1,
G2, G3, G4, G12, G22, G14, G14') collected from the coastal zone show the water type of Na-Cl or
Na•Ca-Cl•SO$_4$ or Ca•Na•Mg-Cl•SO$_4$, indicating that, apart from seawater intrusion, the anthropogenic
pollution also plays important role on modifying the groundwater chemistry.

The seawater from Bohai Sea has relatively higher NO$_3^-$ concentrations (810 mg/L in this study, up to

1092 mg/L in the coastal seawater of Dalian, Han et al., 2015) due to wastewater discharge into the sea.
NO$_3^-$ concentration of groundwater in the well field increased from 5.4 mg/L in May 1985 to 146.8~339.4
mg/L in Aug 2010, while the concentration of seawater in this area changed from 57.4 mg/L in May1985 to
810.1 mg/L in Aug 2010. The diagram (Fig. 7) of Cl$^-$ vs. NO$_3^-$ concentration of groundwater can be used to
identify the different mixing trend in this study area, including the mixing process with contaminated
seawater, and the anthropogenic NO$_3$-sources (e.g. domestic/industrial wastewater discharge, NO$_3^-$-bearing



fertilizer input through precipitation infiltration and the irrigation return-flow) in the inland area. It can be
seen from Fig. 7 that the major source of $NO_3^-$ in groundwater is from anthropogenic input, with the
exception of G10 and G15 mixing with seawater in the well field. The deep groundwater (e.g. G9, G14') is
also contaminated by higher $NO_3^-$ concentrations, which is likely associated with the abandoned wells.
According to one investigation by Zang et al.(2010), 14 of 21 pumping wells in the Zaoyuan well field
have been abandoned due to the salinized water quality, and 307 pumping irrigation wells (occupied 2/3 of
total pumping wells for irrigation) have been abandoned. However, the local department has not made any
measure to deal with those abandoned wells.
5.2 Groundwater salinization processes
*5.2.1 Development of seawater intrusion and associated hydrochemical behavior*
The intensively pumping groundwater from the Quaternary aquifer of Yang-Dai River coastal plain
has resulted in the development of groundwater depression cones from Zaoyuan well field to Fangezhuang,
with the aggravation of seawater intrusion in this region. In the 1950s, the seawater intrusion in the study
area was only occurred within 2 km distance from the coastline, and it expanded to over 5 km distance in
the 1980s. In 1986, the groundwater depression cone centered in the Zaoyuan well field was characterized
by 6 meters depths below the sea level, with the water level -3 m.a.s.l. The enclosed area by 0 m.a.s.l water
level contours covered 10 km$^2$. The original Ca-HCO$_3$ type water changed to Ca•Na-Cl type. Apart from
the intensive exploitation fresh groundwater from coastal aquifer, the successive drought (1976-1989) also
played important roles on controlling the groundwater recharge and exacerbating seawater intrusion in the
coastal area of north China (Wilhite, 1993; Han et al., 2015). In this study area, the annual mean
precipitation was 668.7 mm during 1954-1995, the Cl concentrations was ranging from 130 to 170 mg/L in
the Zaoyuan well field. Whereas the annual mean precipitation reduced to 559.7 mm during 1996-2011,
resulting in Cl concentration in the well field up to 550 mg/L in May 1986, and 812 mg/L in July 2006. It
has seriously threatened the safety of water supply in this region. The seawater intrusion in the coastal
aquifer shows the wedge-shaped body and has vertically characterized by freshwater in the upper part and
salt water in the lower part of shallow aquifer. Since 2002, with the establishment of anti-tide dam in the
Yang River estuary area, it has good effect on preventing the horizontal pouring seawater into riverway.
Thus, the seawater intrusion is mainly caused by lateral inflow of seawater in the aquifer.
According to the guidelines of drinking water standards from China Environmental Protection




Authority (GB 5749-2006) or US EPA or WHO, the guideline of chloride concentration for drinking water
is 250 mg/L. Most groundwater distributed in the seawater intrusion area cannot be used for irrigation, the
source of drinking water and industrial utilization. It has enhanced the scarcity of fresh water resources in
this region by vicious cycle of groundwater level decline → seawater intrusion → groundwater salinization
→ groundwater level decline again. This will also influence the surface water runoff. How to judge the
criterion of seawater intrusion? Generally, 250 mg Cl/L can be regarded as the intruded standard, and more
than 1000 mg Cl/L as the serious intrusion (Jiang and Li, 1997; Zhuang et al., 1999). Some studies took the
TDS (>1000 mg/L) as the intruded standard (e.g., Xue et al., 1997; Zhang and Peng, 1998). Water type can
also be used as the intruded standard, such as Ca-Cl type water occurs during seawater intrusion into
freshwater aquifers, and Na-HCO$_3$ type water displays during flushing of the mixing zone by freshwater
(Appleo and Postma, 1993). Additionally, the multi-hydrochemical ionic ratios can also provide important
confirmation of hydrogeochemical processes modifying groundwater chemistry during seawater intrusion
(Vengosh et al., 1997; Jones et al., 1999). However, the frequent anthropogenic activities modified coastal
hydrologic dynamics and hydrogeochemical characteristics to great extent. For instance, with except of
modern seawater, the sources of chloride in groundwater system could be derived from paleoseawater relics
in aquifers, infiltration of agricultural return flow with fertilizer solutions, and discharge of industrial and
domestic wastewater.

Hydrogeochemical facies evolution diagram (HFE-D) proposed by Giménez-Forcada (2010) can be

used to analyze the phase of seawater intrusion or freshening and its dynamics. From Fig. 8, it can be seen
that most brackish groundwaters (e.g., G11, G16, G17, G20, G25, G28, G29) have evolved in the series of
Ca-HCO$_3$ → Ca-Cl→ Na-Cl facies under the intrusion period. Locally, several water samples (e.g., G1,
G12, G26) collected from the interfluve area have been characterized by freshening process. Deep
groundwater G11 sampled from the well field shows being under salinizing period in the relatively dry
season, and under freshening period in the relatively wet season. In the coastal zone, the river water has
obvious mixing trend between end members. Some shallow groundwaters (e.g., G2, G4, G10, G13, G15)
are close to the mixing line between end members on this figure, indicating significant mixing with
seawater. The groundwaters (G10, G11, G15, G26) collected from the productive wells of the Zaoyuan
well field display the different processes occurring in salinizing or freshening stages, indicating that the
heterogeneous hydrogeological conditions could be responsible for this distinguished patterns.

The calculated results of saturated indices (Fig. 9) show that that SI$_{cal}$ and SI$_{dol}$ have some deviation





from equilibrium (-0.4 to +0.5 for $SI_{cal}$, and -0.5 to +0.5 for $SI_{dol}$). The distribution of $SI_{cal}$ and $SI_{dol}$ is
related to the sampling period. In the wet season, most of water samples are characterized by $SI_{cal}$ <0 and
$SI_{dol}$ <0, suggesting they are under unsaturated for these minerals, while in the dry season, most of water
samples are under saturated with respect to these minerals. In contrast, all sampled groundwater had
negative saturation indices with respect to gypsum ($SI_{gyp}$<0), indicating that these water samples are
under-saturated with respect to gypsum. The plots (Fig. 10 a-f) of ionic molar ratios (Na/Cl, $SO_4$/Cl, Mg/Ca,
Ca/($SO_4$+$HCO_3$), Ca/$SO_4$, and (Ca+Mg)/Cl) can be used to further reveal the groundwater salinized
processes and dominated hydrochemical behavior. The brackish groundwaters in this study area have an
enriched $Ca^{2+}$ (i.e., the ratio of Ca/($HCO_3$+$SO_4$)>1 with low ratios of Na/Cl and $SO_4$/Cl as the seawater
proportion in the mixture increases. As shown in Fig. 10a, Na/Cl ratios of brackish groundwater affected by
seawater intrusion are usually lower than the ratio (0.86) of modern seawater. The high Na/Cl ratios (>1)
could be typical of anthropogenic sources (i.e., domestic waste waters). When seawater intrudes into
coastal freshwater aquifers, $Ca^{2+}$ on the clay-bearing sediments can be replaced by $Na^+$:

$2Na^+ + Ca\text{-}X_2 \rightarrow 2Na\text{-}X + Ca^{2+}$

This process can decrease the Na/Cl ratios and increase the (Ca+Mg)/Cl ratios. The dolomitization process
can be described by the transformation reaction (Appelo and Postma, 2005):

$2CaCO_3 + Mg^{2+} = CaMg(CO_3)_2 + Ca^{2+}$

It can result in Ca-enrichment over Mg in solution and that Mg/Ca ratios decreases. This process may also
be characterized by Ca-Cl water type.

To explain the enrichment in $Ca^{2+}$ relative to $SO_4^{2-}$ concentrations, observed in most water samples

(Fig. 10e), gypsum dissolution ($SI_{gyp}$<0) can be coupled by cation exchange reactions under the interaction
with clay stratum and calcite precipitation with incongruent dissolution of dolomite and gypsum.
Additionally, due to the ORP values ranging from 3 to 74 mV for 18 of 22 water samples collected in
August 2010, the sulfate reduction under anaerobic conditions may be responsible for relatively high
Ca/$SO_4$ and low $SO_4$/Cl ratios. Generally, low Na/Cl, $SO_4$/Cl and high Ca/($HCO_3$+$SO_4$) (>1) ratios are
further indicator of the arrival of seawater intrusion.

∆Na is negative in most samples of this area (Fig. 11a and Fig. 12a). The depletion of $Na^+$ could be

caused by the inverse cation exchange taken place with the clay sediments. This exchange produces Ca
release to the solution during the seawater intrusion. The positive ∆Ca and ∆Mg may be due to the
dissolution of calcite, dolomite and gypsum present in the aquifer strata. Water flushing during aquifer





recharge can result in positive △Na and negative △Ca and/or △Mg (Fig. 11a). For some water samples, the
Ca enrichment is not accompanied by Na depletion, which could be caused by dolomitization (Ca
enrichment with Mg depletion) (Fig. 12b). The excess of $SO_4$ compared to conservative mixing (Fig. 11d)
can be explained by redissolution of the precipitated gypsum along the mixing front.
*5.2.2 Mixing between thermal and cold groundwater*
Sea level rose by about 100 m since the end of the last glacial period (18,000 years, 18 ka BP) and
stabilized around 5 ka BP in the eastern China (Yang, 1996). The marine sediments could not be found in
the geothermal field, indicating the transgression in the geologic history did not occur around Danihe area
(Zeng, 1991). However, the fracture and structural fissure developed well in this study area became the
major subsurface pathway of seawater intrusion. The previous studies have revealed that the geothermal
waters in this area are characterized by the features of residual seawater and modern precipitation (Zeng,
1991; Hui, 2009). The results of $^{14}C$ age dating for the geothermal waters in this area are ranging from 3.4
ka to 12.8 ka with lower tritium contents (less than 2TU) (Zeng, 1991). The Piper plot (Fig. 5) shows
CaNa-Cl type water for the geothermal waters. It is noteworthy that the geothermal water from Danihe
geothermal field has higher Sr concentrations (up to 89.8 mg/L) relative to that in seawater (5.4-6.5 mg/L in
this study), due to the Sr-bearing minerals (i.e., celestite, strontianite) with Sr contents of 300-2000 mg/kg
present in the bedrock (Hebei Geology Survey, 1987). The mixture waters sampled from the geothermal
field in this study also have the higher Sr concentrations relative to seawater, i.e., G9 with Sr concentrations
ranging from 7.4 to 11.6 mg/L, and G19 from 4.9 to 7.1 mg/L. The diagram of chloride versus strontium
concentrations of different water samples (Fig. 13) shows that the groundwater samples (e.g., G9, G19)
collected from the geothermal field have obviously been characterized by closing to mixing line between
fresh cold- and thermal-groundwater. Some waters (G16, G20, G29) sampled from the downstream area
also close to this mixing line, indicating the thermal water overflows into the coastal aquifers in different
depths. The water samples collected from the well field are located between two mixing lines (Fig. 13),
suggesting the groundwater in the well field simultaneously suffered from the mixing with thermal water
and obvious seawater intrusion. Additionally, the points of water samples (G5, G8, G9, and G19), collected
from the geothermal field, mainly occurs on the HFE-D (Fig. 8) in the *12* (MixCa-Cl) and *16* (Ca-Cl) facies
zones, indicating these waters have been modified by the reverse base-exchange reactions.
As both Cl and Br are not affected by water-rock interactions and usually behave conservatively, the
Cl/Br ratio can be used as a reliable tracer to study the processes of evaporation and salinization of water
(Edmunds, 1996; Jones et al., 1999). Standard seawater (Cl/Br molar ratio=650.8) may be distinguished
from relics of evaporated seawater (normally less than 669.3), input of evaporite dissolution (more than
2256) and anthropogenic pollution (e.g., sewage effluents, Cl/Br ratios up to 1805; Vengosh and Pankratov,
1998) or agricultural return-flows with low Cl/Br ratios (Jones et al., 1999). It can be seen from Fig. 14 that
the points of the thermal waters lie lower than the ratio line of standard seawater, indicating that they are
affected by mixing with relics of evaporated seawater. The points of cold groundwaters (G9, G19) sampled
from the geothermal field display between the seawater and the thermal waters, indicating these cold waters
are mixture between cold groundwater and the thermal water, which has relics of evaporated seawater.
However, it cannot exclude adding the Br inputs into groundwater system through the pesticides
application of the pronounced agricultural activity (Davis et al., 1998), this effect could lower the Cl/Br
ratios of the groundwaters. The groundwater sample G10 in the well field shows the feature of high Cl/Br
ratio in Fig. 14, indicating obvious anthropogenic inputs (e.g., discharge domestic wastewater) occurring in
the shallow aquifers around the well field.
*5.2.3 Interaction between surface- and ground-water*
Coastal zones encompass the complex interaction among different waters (i.e., river water, seawater,
groundwater, rainfall water). The interaction between surface- and ground-water in the Yang-Dai River
coastal plain is usually ignored by the previous studies. However, understanding how surface water
interacts with the groundwater is essential for managing freshwater resources. Groundwater depression
cone below the sea level has formed in the early 1980s. Due to the irrigation supported by transfer of
surface water from the upper and middle stream of Yang-Dai River, the amount of surface water discharged
into the Bohai Sea declined to great extent. Under the tide effects, seawater can be poured into the estuary
of the downstream section of the rivers, resulting in the river bed filled with saltwater, which can cause
mixing between river water and seawater. The results of water chemistry analysis from two river sections
show that the distribution of salt water reached more than 10 km above the estuary of the Yang River, and
about 4 km above the estuary of the Dai River (Han, 1988). It leaded to that the seawater simultaneously
intruded into the coastal aquifers through not only the lateral subsurface flow from coastline to the inland
but also vertical infiltration from the riverbed to both sides of the river. The hazard caused by the latter
pattern had been more serious than the former pattern, before the establishment of anti-tide dam at the





estuary of Yang River. Currently, the seawater-intruded distance towards inland has been controlled within
4 km away from the coastline.

The stable isotope compositions of different water samples (Fig. 4) display that the points of most

surface water samples are deviated from the LMWL to the right, indicating that these waters are likely to be
subject to evaporation to different degrees. The points of surface water samples (S1, S2, S3, and S7) in
Fig.4 close to the compositions of local seawater, indicating the pronounced mixing process with seawater
for these surface waters. However, the samples S2, S6, S8, and S9 have the depleted compositions of stable
isotopes, probably resulting from the exchange between them and local groundwater. S12, located at the
estuary area, has variable compositions due to the sampling seasons. HFE-D shows that most of surface
water samples are close to the mixing line between end members (freshwater and seawater). S9 is
significantly characterized by salinization process probably due to the interaction with ambient
groundwater. It can be seen from the relationship between ionic delta values and seawater proportions for
the water samples (Fig. 11) that G1, G11, G12, G14', G26, G29, due to these wells close to the river or
located at the flat interfluve, may be dominated by the obvious freshening process. While G2, G3, and G16
under the salinizing process could be subject to the vertical infiltration of saltwater in the river. The points
of surface waters (S1, S2, S3, and S7) on Fig. 4 and Fig. 8 are distributed along the mixing line between
fresh- and sea-water end members. It is due to the direct mixture occurs in the riverway. S12 sampled from
the Dai River estuary may be contaminated by the wastewater discharge with higher Sr concentration
relative to seawater. By contrast, surface water from the Dai River have higher seawater proportions
compared with that from Yang River, owing to that the local government did not establish anti-tide dam in
the Dai River estuary. G1, G2, G3, G10 and G13 collected from coastal zone are obviously mixed with
seawater with closing to the mixing line between seawater and freshwater in Fig. 13.
### 5.3 Conceptual model of groundwater flow patterns

Generally, groundwater in this study area is mainly originated from precipitation, river infiltration,

lateral subsurface runoff, upflow of geothermal waters and seawater intrusion in the coastal area. The
associated hydrological processes driven by the natural (hydrologic, geologic, climatic) changes and
anthropogenic activities have resulted in groundwater salinization processes, along with the complex
hydrogeochemical characteristics of groundwater system. Groundwater changes from Ca-HCO$_3$ type water
in the piedmont area to the Na-Cl type water in the coastal area.





A conceptual model groundwater flow system in the Yang-Dai River coastal plain can be summarized
in Fig. 15. Four subsurface processes, including seawater intrusion, return-flow of agricultural irrigation,
mixing with geothermal water and interaction between surface water and groundwater/seawater, could be
responsible for the groundwater salinization in this area. Two aspects of seawater intrusion, identified by
depleted $\Delta$Na and enriched $\Delta$Ca with Ca-Cl type water and Ca/(HCO$_3$+SO$_4$)>1 and lower Na/Cl and
SO$_4$/Cl relative to these ratios of seawater, can be delineated, namely vertical infiltration of saltwater inflow
towards the inland estuary and lateral inflow of seawater driven by over-pumping groundwater from fresh
aquifers. Irrigation return-flow from local groundwater can cause groundwater nitrate pollution (up to 340
mg/L NO$_3^-$ in groundwater of this area) due to the infiltration and dissolution of fertilizers. Geothermal
water with high TDS, F, and Sr concentrations flows into the Quaternary aquifers, mixing with cold
groundwater, and transports to the downstream area of Yang River Basin. Additionally, the interaction
between surface- and ground-water can cause seasonal flushing local groundwater in the upstream
interfluve or lead to saltwater infiltration affected by tide/surge along the riverbed at the estuary.
**6. Conclusions**
It has been recognized that groundwater in the Quaternary aquifers of the Yang-Dai River coastal plain
is the important water resource for agricultural irrigation, urban and tourism development and industrial
utilization. Natural climate change (e.g., continuous drought, overflow of geothermal water) and human
activities have made the problem of groundwater salinization in this area increasingly prominent, even
resulting in the closure of the Zaoyuan well field. Based on the analysis of hydrochemical and stable
isotopic compositions of different water bodies, including surface water, cold groundwater, geothermal
water, and seawater, we delineated the groundwater flow system and groundwater salinization processes.
Seawater intrusion is the main aspect responsible for the groundwater salinization in the coastal zone,
including the vertical saltwater infiltration along the riverbed into aquifers, which is affected by the
tide/surge process, and the lateral seawater intrusion caused by pumping for fresh groundwater. The
overflow of the high mineralized thermal water into the Quaternary aquifers along the fault zone mixes
with the cold groundwater and makes it salinized. The thermal water has characterized by lower Cl/Br
ratios and higher Sr concentrations relative to seawater. It cannot be ignored that the salinization or nitrate
pollution from the anthropogenic activities (e.g., agricultural irrigation return-flow with solution of
fertilizers). Additionally, the interaction between surface- and ground-water can also affect the groundwater





salinization in this area. Different approaches of hydrochemical analysis, such as Piper plot, HFE-D, major
ionic ratios (Na/Cl, $SO_4$/Cl, Ca/$SO_4$, (Ca+Mg)/Cl, Ca/($SO_4$+$HCO_3$), Cl/Br) and Sr, were used in this study
to identify the different hydrogeochemical reactions and freshening or salinizing processes in the
Quaternary aquifers.

Groundwater salinization has become a prominent water environment problem in the coastal area of

north China, which has caused the further paucity of fresh water resources and has become the bottleneck
of urban development to a certain extent. Since the 1990s, the local government has begun to pay attention
to the development of seawater intrusion, and the irrational exploitation has been restricted. The Zaoyuan
well field has ceased to pump groundwater since 2007. The anti-tide dam has been established in the Yang
River estuary area in 2002, effectively intercepting the seawater pouring into riverway during the tide/surge
period. These actions have made the rate of intrusion slowed down. The joint use of surface water and
groundwater with reasonable exploitation program is essential and economical for the local water resources
management. However, the quantitative understanding to the vertical and lateral saltwater intrusion into
fresh aquifers should be obtained from further continuous groundwater monitoring and numerical
groundwater flow and transport modeling. This study would be benefit the local agricultural development
and groundwater resources management.
**Acknowledgement**

This research was supported by Zhu Kezhen Outstanding Young Scholars Program in the Institute of
Geographic Sciences and Natural Resources Research, Chinese Academy of Sciences (CAS) (grant number
2015RC102), and Outstanding member program of the Youth Innovation Promotion Association, CAS
(grant number 2012040). The authors appreciate the helpful field work and data collection made by Mr
Yang Jilong from Tianjin Institute of Geology and Mineral Resources, Dr. Wang Peng and Dr Liu Xin from
Chinese Academy of Sciences.

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

with English abstract)



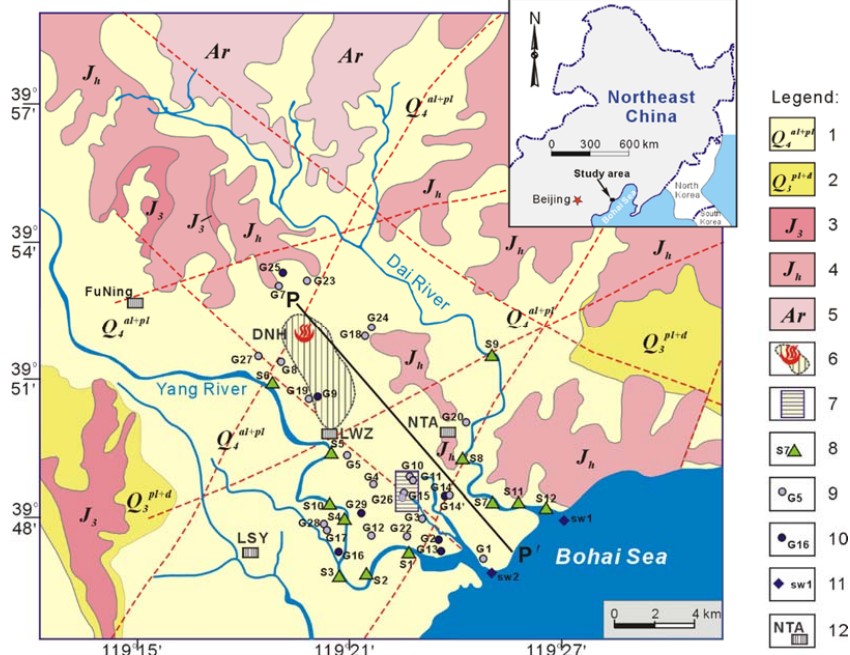

Figure 1. Map for showing geological background and water sampling sites in the study area

Explanation for the legend: 1-Holocene (al- alluvial, pl- pluvial) sediments; 2- Upper Pleistocene (dpl-proluvial-deluvial) sediments; 3- Jurassic andesite; 4- Jurassic migmatitic granite; 5- Archaean group gneiss; 6-Geothermal field location and its influence area; 7- Zaoyuan well field; 8- surface water sampling site; 9-shallow groundwater sampling site; 10- deep groundwater sampling site; 11- seawater sampling site; 12-village/county site (DNH-Danihe; NTA-Niutouai; LWZ-Luwangzhuang; LSY-Liushouying). The red dashed lines are showing the buried fault distribution in this area.

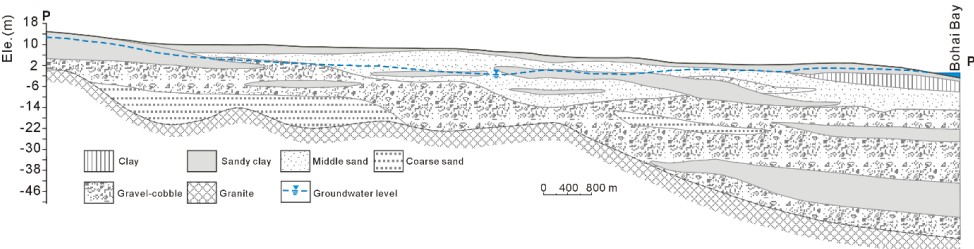

Figure 2. Hydrogeological cross-section of Yang-Dai River Plain (P-P′ in Fig. 1) (modified from Han, 1988)





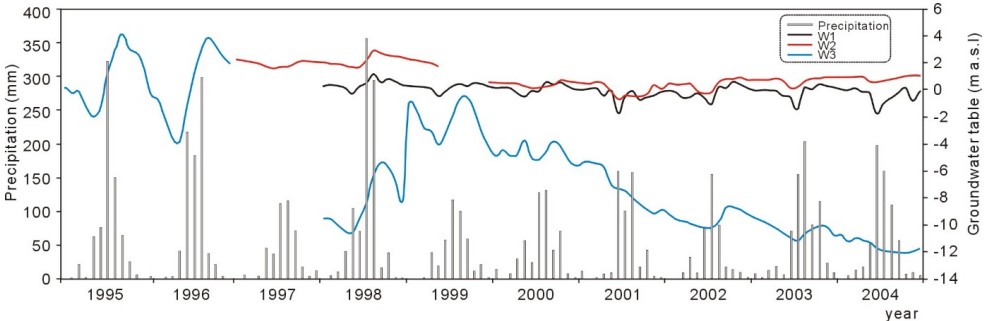

Figure 3. Distribution of precipitation and dynamics of groundwater table in the study area

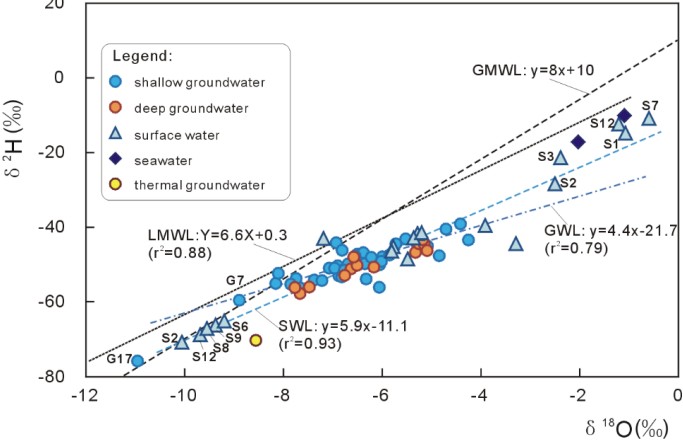

Figure 4. Stable isotope compositions of different water samples collected from the study area

LMWL - local meteoric water line; GMWL – global meteoric water line (Craig, 1961); GWL – groundwater
line; SWL – surface (river) water line.

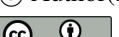


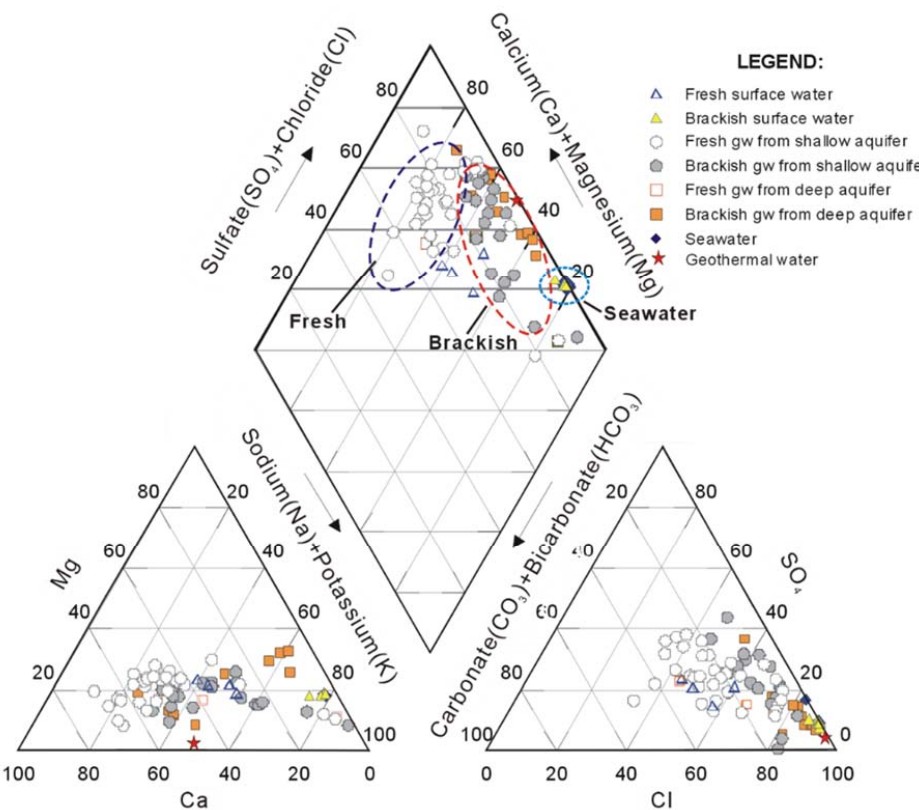

Figure 5. Piper plot of different water samples

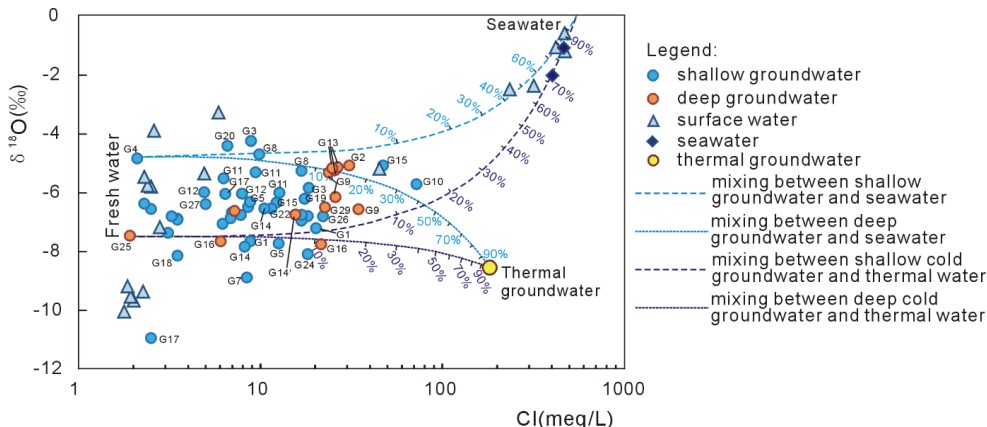

Figure 6. Relationship between chloride content and isotopic signature of different water samples as a means to differentiate mixing processes in the area. The data of thermal water are from Zeng (1991).




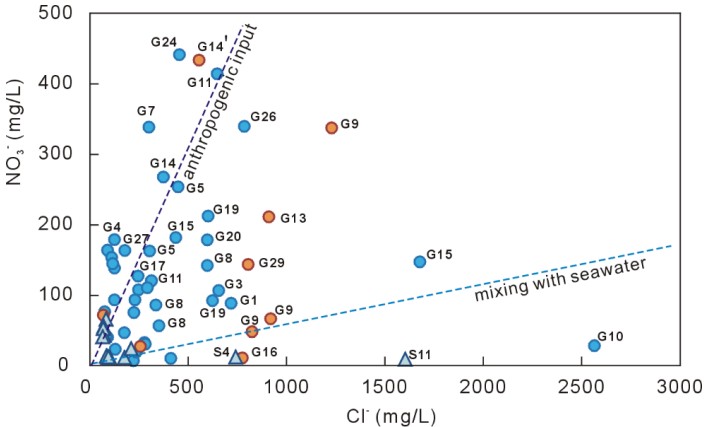

Figure 7. Plots of chloride versus nitrate concentrations. Symbols are same as in Fig. 6.

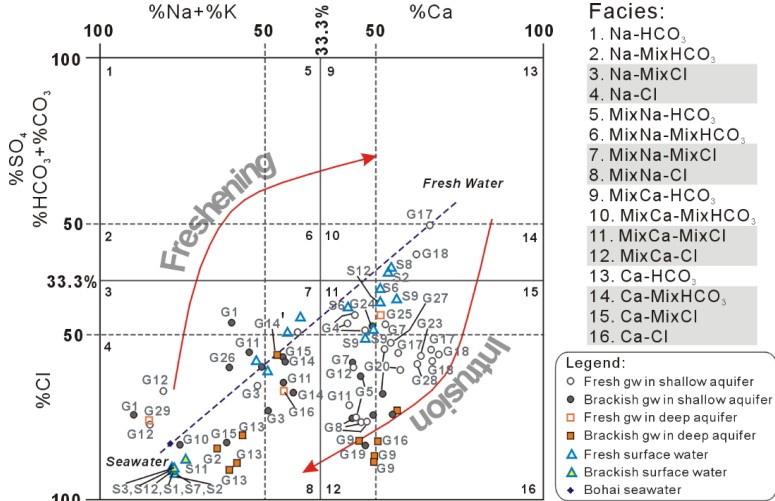

Figure 8. Hydrogeochemical facies evolution diagram (HFE-D) for the collected water samples





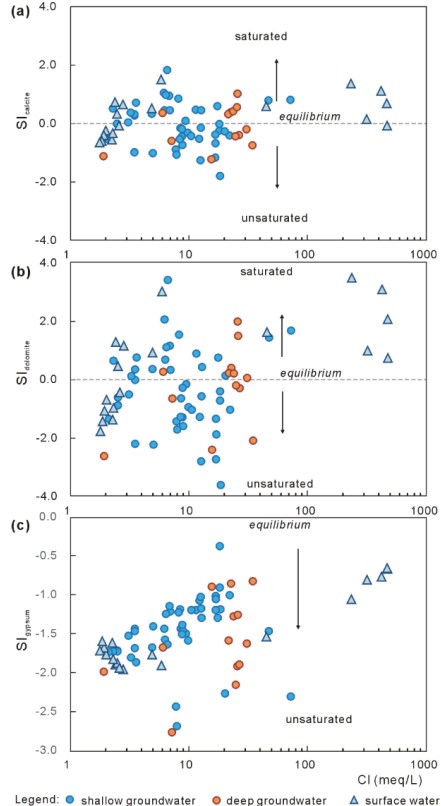

Figure 9. Variation of saturation indices with respects to calcite (a), dolomite (b) and gypsum (c) versus chloride concentrations





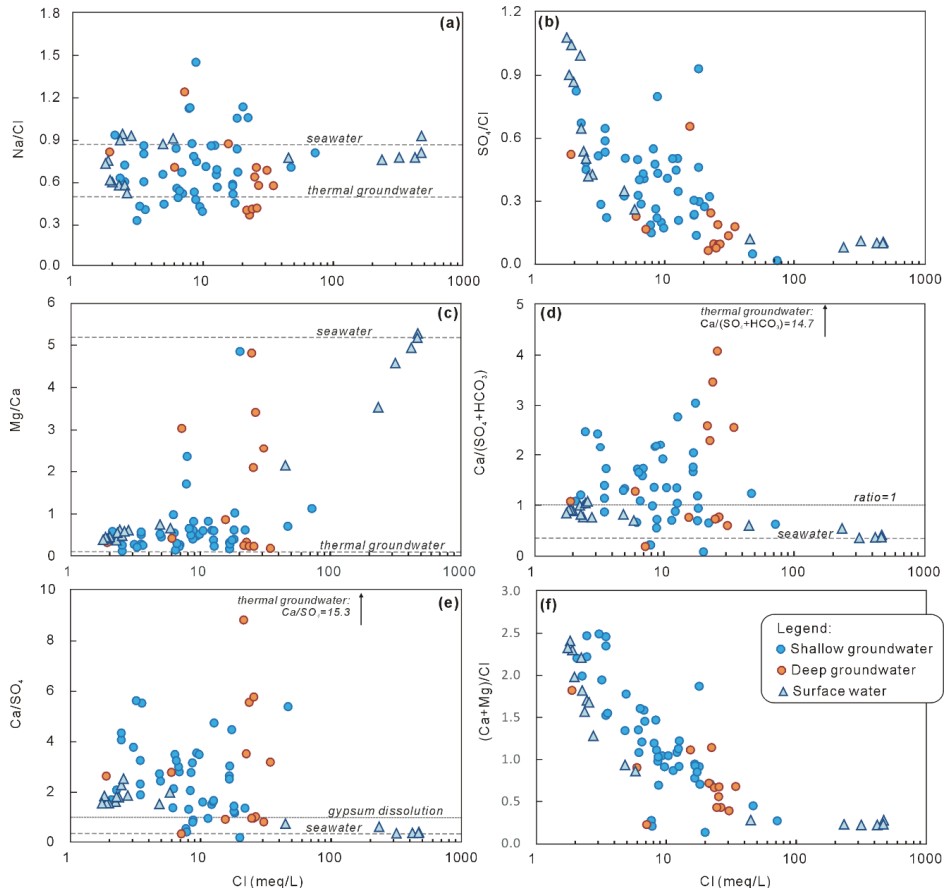

Figure 10. Molar ratios of major ions versus chloride concentrations for different water samples from the study area





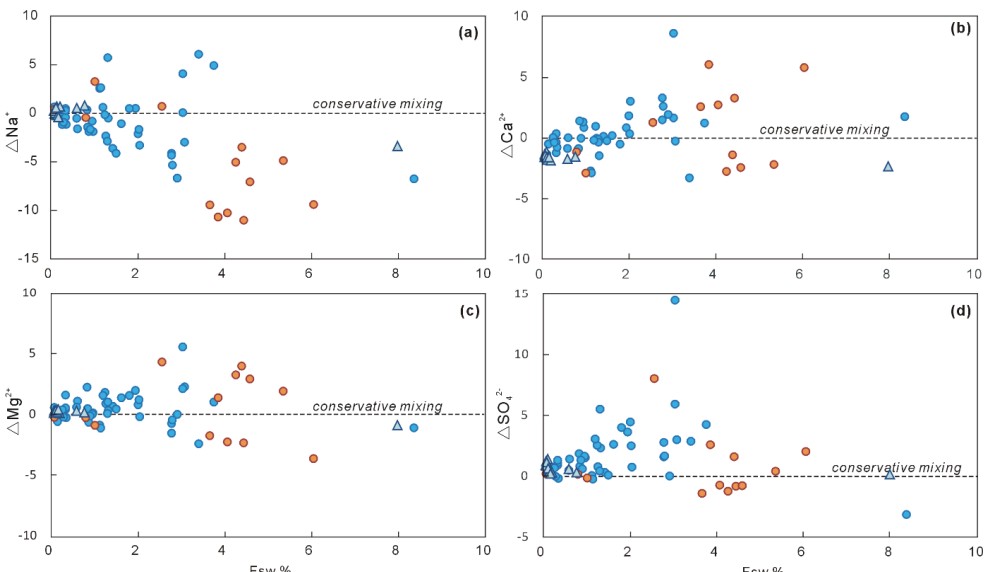

Figure 11. $\Delta Na^+$ (a), $\Delta Ca^{2+}$ (b), $\Delta Mg^{2+}$ (c) and $\Delta SO_4^{2-}$ (d) versus calculated seawater percentage ($F_{sw}$%)

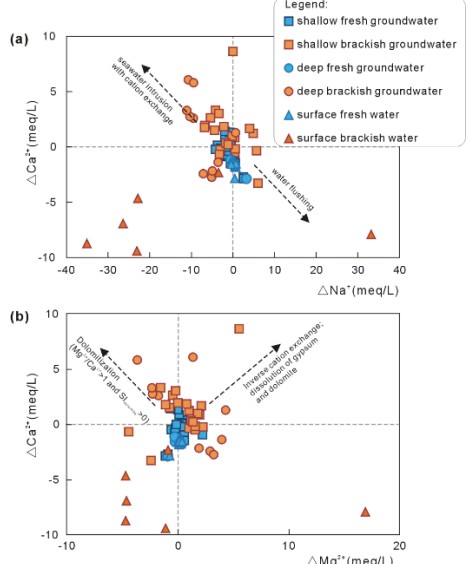

Figure 12. Distribution of cationic deltas (a)- $\Delta Na^+$ versus $\Delta Ca^{2+}$; (b) $\Delta Mg^{2+}$ versus $\Delta Ca^{2+}$ for each sample
(in meq/L)

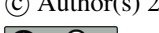

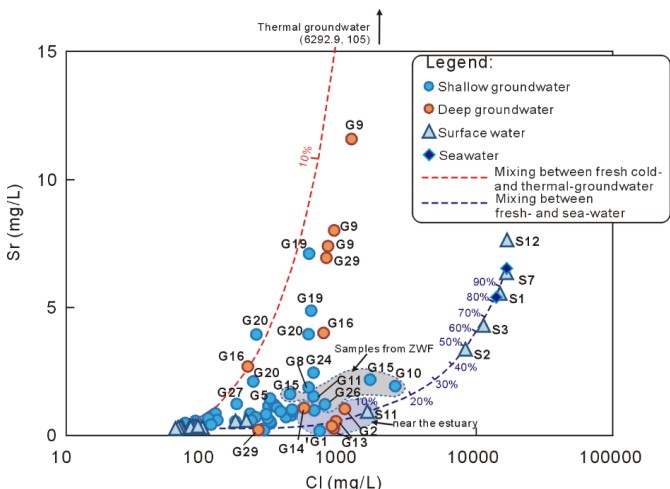

Figure 13. Chloride versus strontium concentrations of different water samples
ZWF- Zaoyuan well field

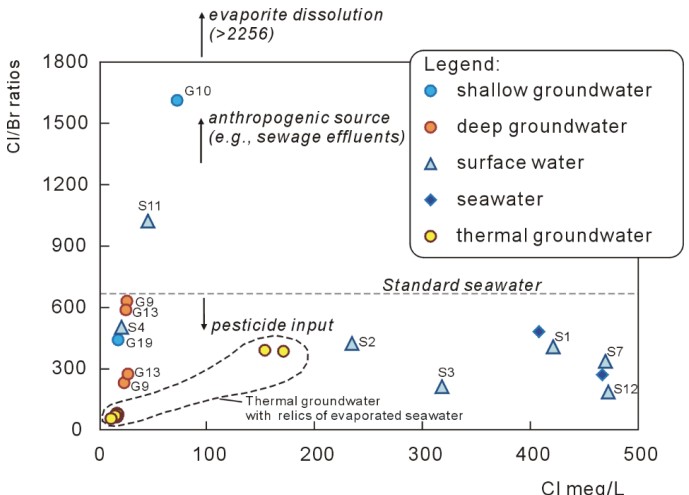

Figure 14. Chloride versus bromide to chloride molar ratios. Data of 8 thermal groundwater samples are from Zeng (1991) and Hui (2002).



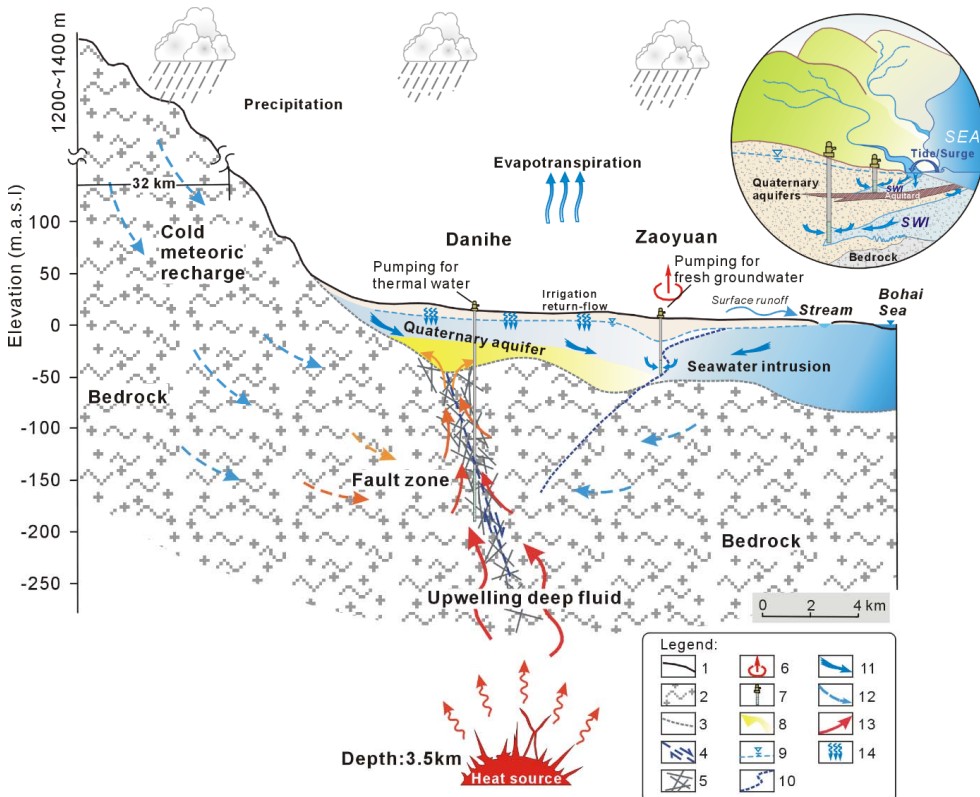

Figure 15. Conceptual model of groundwater flow system in the Yang-Dai River coastal plain

Explanation: 1- Land surface; 2- Bedrock; 3- Boundary between Quaternary sediments and bedrock; 4-Fault; 5- Permeable fracture zone; 6- Concentrated groundwater pumping zone; 7- Pumping wells; 8- Zone affected by upflow of geothermal fluids; 9- Groundwater table; 10- Potential interface between fresh- and salt-water; 11-Groundwater flow direction in Quaternary aquifers; 12- Groundwater flow in bedrocks; 13- Geothermal groundwater flow direction; 14- Irrigation return-flow.





Table 1. Physical, hydrochemical and isotopic data of the water samples collected from the Yang-Dai River coastal plain

| WaterType | ID | Sampling Time | Ele. m | WellDepth m | WaterTable Depth(m) | EC μs/cm | pH | T °C | ORP mV | DO mg/L | Cl⁻ mg/L | NO₃⁻ mg/L | SO₄²⁻ mg/L | HCO₃⁻ mg/L | Ca²⁺ mg/L | Na⁺ mg/L | K⁺ mg/L | Mg²⁺ mg/L | Sr mg/L | δ²H ‰ | δ¹⁸O ‰ |
|---|---|---|---|---|---|---|---|---|---|---|---|---|---|---|---|---|---|---|---|---|---|
| *Shallow groundwater samples:* | | | | | | | | | | | | | | | | | | | | | |
| Fresh Groundwater | G4 | Aug.2010 | 5 | 15 | 4.2 | 741 | 6.5 | 20.5 | 6 | 3.9 | 124.3 | 92.6 | 89.7 | 136.9 | 70.3 | 69.2 | 2.3 | 21.8 | 0.67 | -50 | -6.9 |
| | G27 | Aug.2010 | 3 | 12 | | 1014 | 6.4 | 14.8 | 16 | 3.8 | 177.5 | 163.0 | 121.0 | 128.0 | 122.0 | 50.6 | 2.6 | 33.4 | 1.22 | -50 | -6.4 |
| | G12 | Aug.2010 | 8 | 10 | | 1152 | 7.2 | 25.0 | 9 | 5.9 | 276.9 | 31.9 | 69.8 | 148.8 | 15.8 | 201.9 | 11.1 | 16.2 | 0.25 | -51 | -6.8 |
| | G17 | Aug.2010 | 5 | 25 | 5.7 | 624 | 7.1 | 19.1 | 13 | 3.3 | 88.8 | 39.5 | 58.3 | 223.3 | 98.4 | 41.4 | 4.3 | 7.5 | 0.33 | -76 | -11.0 |
| | G18 | Aug.2010 | 5 | 23 | 1.7 | 934 | 7.4 | 14.9 | 12 | 7.1 | 124.3 | 137.8 | 98.7 | 235.2 | 133.3 | 48.3 | 2.2 | 23.2 | 0.71 | -55 | -8.2 |
| | G1 | Sep.2009 | 6 | 8 | 2.4 | 1673 | 8.3 | 19.4 | | | 220.1 | 2.0 | 148.3 | 207.9 | 84.3 | 119.6 | 17.1 | 49.9 | 0.66 | -51 | -7.1 |
| | G4 | Sep.2009 | 6 | 15 | 4.6 | 1295 | 7.9 | 13.9 | | | 124.3 | 178.5 | 108.7 | 92.4 | 104.4 | 64.4 | 2.7 | 35.9 | 0.85 | -44 | -6.9 |
| | G5 | Sep.2009 | 11 | 16 | 6.5 | 1544 | 7.8 | 13.3 | | | 303.4 | 162.3 | 108.2 | 38.5 | 124.5 | 103.7 | 1.1 | 39.1 | 1.14 | -47 | -6.5 |
| | G23 | Sep.2009 | 15 | 30 | 3.0 | 556 | 7.9 | 14.9 | | | 88.8 | 163.4 | 54.0 | 51.9 | 97.4 | 34.5 | 0.5 | 15.6 | 0.55 | -48 | -6.6 |
| | G7 | Sep.2009 | 17 | 8 | 1.7 | 901 | 8.4 | 14.9 | | | 81.7 | 41.2 | 74.2 | 69.3 | 64.0 | 33.2 | 1.0 | 16.5 | 0.41 | -47 | -6.4 |
| | G8 | Sep.2009 | 4 | 13 | 4.3 | 1621 | 7.8 | 14.0 | | | 334.6 | 85.4 | 90.1 | 69.3 | 132.5 | 91.9 | 1.4 | 38.5 | 1.19 | -44 | -5.3 |
| | G11 | Sep.2009 | 4 | 13 | | 1237 | 8.6 | 18.4 | | | 222.9 | 74.5 | 98.7 | 30.8 | 87.4 | 80.1 | 1.8 | 29.0 | 0.84 | -43 | -5.5 |
| | G12 | Sep.2009 | 8 | 10 | | 1114 | 7.9 | 15.0 | | | 174.0 | 46.0 | 76.4 | 107.8 | 86.2 | 73.6 | 4.3 | 27.2 | 0.53 | -48 | -6.0 |
| | G28 | Sep.2009 | 6 | 23 | 6.8 | 1748 | 8.2 | 14.6 | | | 127.8 | 22.5 | 38.4 | 107.8 | 88.2 | 33.3 | 1.3 | 14.0 | 0.58 | | |
| | G18 | Sep.2009 | 9 | 23 | 5.5 | 2850 | 7.8 | 13.7 | | | 110.1 | 152.9 | 76.7 | 53.9 | 120.3 | 23.1 | 1.1 | 20.5 | 0.65 | -54 | -7.4 |
| | G20 | Sep.2009 | 12 | 30 | | 1602 | 7.7 | 19.5 | | | 246.9 | 107.1 | 135.9 | 107.8 | 158.2 | 82.8 | 1.7 | 26.3 | 3.94 | -50 | -6.7 |
| | G17 | Sep.2009 | 6 | 33 | 7.9 | 819 | 8.1 | 14.3 | | | 243.5 | 126.4 | 141.7 | 161.7 | 176.3 | 105.5 | 1.8 | 24.6 | 0.71 | -53 | -6.9 |
| | G3 | Jun.2008 | 4 | 20 | | 1573 | 7.3 | 14.5 | | | 315.0 | | 184.4 | 54.9 | 67.3 | 152.1 | 3.8 | 33.2 | 0.46 | -43 | -4.2 |
| | G4 | Jun.2008 | 6 | 6 | | 688 | 7.5 | 14.1 | | | 74.6 | 75.8 | 83.2 | 60.4 | 58.2 | 45.2 | 1.2 | 20.6 | 0.47 | -48 | -4.8 |
| | G5 | Jun.2008 | 5 | 11 | | 1455 | 7.4 | 14.7 | | | 310.9 | 119.5 | 92.7 | 52.2 | 121.3 | 95.4 | 0.9 | 20.9 | 1.43 | -54 | -6.3 |
| | G8 | Jun.2008 | 4 | 30 | | 1402 | 7.3 | 14.7 | | | 349.7 | 55.9 | 81.5 | 85.1 | 118.0 | 88.1 | 1.2 | 36.9 | 1.08 | -40 | -4.7 |
| | G12 | Jun.2008 | 8 | 10 | | 1285 | 7.6 | 19.3 | | | 281.0 | 29.8 | 56.8 | 74.1 | 9.7 | 205.7 | 11.0 | 13.7 | 0.19 | -50 | -6.0 |
| | G17 | Jun.2008 | 6 | 33 | | 1462 | 8.1 | 15.0 | | | 227.9 | 92.6 | 123.8 | 175.7 | 179.2 | 72.1 | 1.1 | 15.9 | 0.57 | -56 | -6.1 |
| | G18 | Jun.2008 | 9 | 23 | | 1125 | 7.9 | 18.8 | | | 115.5 | 144.2 | 44.4 | 90.6 | 103.7 | 31.8 | 1.4 | 13.7 | 0.41 | -53 | -6.8 |
| | G20 | Jun.2008 | 12 | 30 | | 1210 | 9.1 | 28.3 | | | 234.3 | 18.2 | 90.6 | 230.6 | 122.0 | 81.4 | 2.6 | 22.4 | 2.10 | -39 | -4.4 |
| Brackish Groundwater | G1 | Aug.2010 | 6 | 11 | | 2750 | 7.4 | 17.3 | 143 | 1.5 | 312.4 | 120.0 | 337.1 | 205.4 | 112.0 | 294.4 | 29.9 | 41.3 | 0.54 | -56 | -7.7 |
| | G3 | Aug.2010 | 4 | 8 | | 2490 | 5.9 | 14.7 | 74 | 1.4 | 654.3 | 106.0 | 263.7 | 86.3 | 128.2 | 283.6 | 6.6 | 78.6 | 0.97 | -47 | -5.8 |
| | G15 | Aug.2010 | 4 | 30 | | 2290 | 6.6 | 15.4 | 3 | 3.3 | 435.7 | 181.3 | 263.7 | 244.1 | 164.3 | 242.8 | 6.6 | 60.8 | 1.60 | -50 | -6.3 |
| | G26 | Aug.2010 | 4 | 18 | 1.8 | 4060 | 6.4 | 15.2 | 145 | 1.3 | 784.6 | 339.4 | 341.4 | 485.2 | 192.3 | 538.2 | 36.0 | 72.1 | 1.20 | -46 | -6.8 |
| | G11 | Aug.2010 | 6 | 13 | 2.3 | 3490 | 6.5 | 15.0 | 47 | 1.3 | 646.1 | 414.1 | 402.7 | 402.6 | 204.8 | 441.8 | 34.9 | 76.3 | 1.51 | -52 | -6.8 |
| | G20 | Aug.2010 | 12 | 30 | | 1513 | 6.7 | 18.8 | 4 | 4.3 | 596.4 | 177.9 | 245.0 | 184.6 | 268.6 | 225.4 | 2.0 | 29.2 | 3.95 | -51 | -7.0 |
| | G5 | Aug.2010 | 5 | 16 | 5.4 | 1500 | 6.2 | 15.6 | 7 | 5.9 | 447.3 | 253.3 | 304.7 | 79.3 | 204.1 | 188.6 | 0.8 | 47.5 | 0.78 | -54 | -7.7 |
| | G8 | Aug.2010 | 4 | 30 | 2.1 | 1563 | 6.3 | 15.7 | 120 | 4.9 | 596.4 | 141.4 | 188.1 | 104.2 | 196.3 | 220.8 | 3.2 | 38.7 | 1.87 | -44 | -5.3 |
| | G7 | Aug.2010 | 5 | 8 | 0.9 | 1438 | 6.8 | 19.3 | 24 | 6.1 | 299.6 | 338.3 | 192.7 | 128.0 | 164.2 | 151.8 | 0.3 | 50.0 | 0.66 | -60 | -8.9 |
| | G19 | Aug.2010 | 8 | 11 | 2.8 | 2380 | 6.5 | 16.1 | 42 | 3.2 | 600.0 | 211.9 | 192.1 | 119.1 | 241.4 | 199.5 | 6.6 | 42.4 | 7.10 | -52 | -6.8 |
| | G24 | Aug.2010 | 4 | 21 | | 4400 | 6.8 | 15.5 | 4 | 5.9 | 646.1 | 952.1 | 813.3 | 223.3 | 484.1 | 349.6 | 3.4 | 117.4 | 2.44 | -52 | -8.1 |
| | G22 | Sep.2009 | 2 | 20 | 2.0 | 2170 | 8.0 | 17.8 | | | 408.3 | 2.0 | 277.5 | 130.9 | 109.2 | 226.8 | 21.3 | 54.2 | 0.70 | -48 | -6.5 |
| | G10 | Sep.2009 | 3 | 18 | | 9770 | 7.5 | 15.1 | | | 2563.1 | 27.7 | 25.3 | 870.2 | 181.4 | 1340.0 | 98.1 | 123.0 | 1.92 | -44 | -5.7 |
| | G14 | Sep.2009 | 2 | 8.5 | | 1886 | 8.5 | 16.6 | | | 291.1 | 109.8 | 216.4 | 92.4 | 117.8 | 164.7 | 19.3 | 46.5 | 0.82 | -55 | -7.9 |




**Deep Groundwater samples:**

| Sample | Date | n | Depth | c | EC | pH | T | n2 | c2 | TDS | 1 | 2 | 3 | 4 | 5 | 6 | 7 | 8 | δD | δ18O |
|---|---|---|---|---|---|---|---|---|---|---|---|---|---|---|---|---|---|---|---|---|
| G24 | | 11 | 21 | 5.3 | 1003 | 7.8 | 13.6 | 41 | 4.4 | 454.4 | 441.5 | 128.1 | 115.5 | 252.3 | 165.6 | 0.6 | 36.2 | 0.93 | -48 | -6.2 |
| G19 | Sep.2009 | 12 | 11 | 4.7 | 2560 | 8.1 | 14.6 | | | 622.0 | 91.5 | 115.6 | 69.3 | 214.9 | 180.7 | 6.0 | 49.4 | 4.87 | -54 | -7.2 |
| G1 | Jun.2008 | 6 | 8 | | 3150 | 8.5 | 15.0 | | | 717.1 | 87.7 | 265.6 | 98.8 | 9.1 | 527.4 | 26.7 | 26.6 | 0.17 | -49 | -6.0 |
| G11 | Jun.2008 | 6 | 13 | | 2370 | 7.4 | 12.2 | | | 451.2 | | 211.3 | 162.0 | 145.1 | 201.2 | 22.0 | 52.6 | 1.00 | -49 | -6.5 |
| G14 | Jun.2008 | 3 | 8.5 | | 2050 | 7.4 | 18.6 | | | 372.8 | 267.3 | 206.1 | 49.4 | 136.4 | 171.3 | 12.7 | 49.5 | 0.93 | -48 | -6.5 |
| G15 | Jun.2008 | 4 | 18 | | 5990 | 7.5 | 15.6 | | | 1675.6 | 146.8 | 110.1 | 474.9 | 246.5 | 764.7 | 20.2 | 104.9 | 2.17 | -45 | -5.1 |

**Fresh Groundwater**

| Sample | Date | n | Depth | c | EC | pH | T | n2 | c2 | TDS | 1 | 2 | 3 | 4 | 5 | 6 | 7 | 8 | δD | δ18O |
|---|---|---|---|---|---|---|---|---|---|---|---|---|---|---|---|---|---|---|---|---|---|
| G25 | Aug.2010 | 4 | 95 | | 444 | 6.6 | 23.5 | 41 | 5.6 | 68.1 | 71.0 | 48.2 | 89.3 | 52.5 | 35.9 | 1.1 | 10.4 | 0.25 | -56 | -7.5 |
| G16 | Sep.2009 | 3 | 110 | | 1214 | 7.9 | 19.9 | | | 214.4 | | 66.1 | 100.1 | 76.3 | 97.8 | 2.8 | 19.5 | 2.68 | -58 | -7.7 |
| G29 | Sep.2009 | 6 | 60 | | 1291 | 8.1 | 20.9 | 12 | | 255.6 | 26.3 | 57.5 | 69.3 | 8.0 | 205.9 | 12.3 | 14.6 | 0.20 | -51 | -6.6 |
| G29 | Aug.2010 | 6 | 60 | 6.2 | 3220 | 7.2 | 24.1 | 16 | | 803.4 | 143.0 | 264.7 | 178.6 | 386.8 | 189.3 | 4.3 | 77.6 | 6.95 | -50 | -6.5 |
| G16 | Aug.2010 | 3 | 110 | | 1733 | 7.3 | 21.5 | | 4.8 | 766.8 | 5.5 | 67.1 | 205.4 | 246.2 | 197.8 | 4.5 | 37.9 | 4.00 | -56 | -7.8 |
| G9 | Jun.2008 | 9 | 104 | | 3110 | 7.8 | 15.0 | | | 823.6 | 47.4 | 110.5 | 85.1 | 255.1 | 222.2 | 5.1 | 36.7 | 7.40 | -47 | -5.3 |

**Brackish Groundwater**

| Sample | Date | n | Depth | c | EC | pH | T | n2 | c2 | TDS | 1 | 2 | 3 | 4 | 5 | 6 | 7 | 8 | δD | δ18O |
|---|---|---|---|---|---|---|---|---|---|---|---|---|---|---|---|---|---|---|---|---|---|
| G9 | Sep.2009 | 9 | 104 | | 3190 | 8.5 | 20.4 | | | 917.4 | 65.8 | 116.6 | 61.6 | 279.8 | 245.1 | 5.7 | 40.6 | 8.02 | -51 | -6.2 |
| G9 | Aug.2010 | 9 | 104 | | 4600 | 6.3 | 24.3 | 18 | 4.0 | 1228.3 | 337.4 | 296.3 | 92.3 | 392.2 | 455.4 | 5.5 | 45.3 | 11.59 | -48 | -6.6 |
| G14' | Aug.2010 | 5 | 110 | 2.1 | 2850 | 6.1 | 22.9 | 120 | 1.8 | 553.8 | 433.7 | 491.4 | 134.0 | 186.0 | 312.8 | 52.3 | 96.6 | 1.06 | -53 | -6.8 |
| G13 | Aug.2010 | 3 | 90 | | 3230 | 8.9 | 19.2 | 36 | 2.4 | 908.8 | 210.6 | 231.8 | 92.3 | 92.0 | 413.5 | 29.0 | 115.7 | 0.26 | -45 | -5.2 |
| G13 | Jun.2008 | 3 | 90 | | 3180 | 7.9 | 13.5 | | | 945.4 | | 122.2 | 52.2 | 51.4 | 351.9 | 25.0 | 105.4 | 0.55 | -45 | -5.1 |
| G13 | Sep.2009 | 3 | 90 | 2.6 | 3070 | 8.1 | 15.3 | | | 882.0 | | 91.9 | 38.5 | 36.5 | 362.9 | 26.0 | 105.3 | 0.35 | -43 | -5.2 |
| G2 | Jun.2008 | 2 | 60 | | 3780 | 7.7 | 18.5 | | | 1093.4 | | 200.3 | 96.1 | 67.3 | 484.1 | 25.8 | 103.1 | 1.03 | -46 | -5.1 |

**River water samples:**

**Fresh water samples**

| Water body | Sample | Date | c | EC | pH | T | n2 | c2 | TDS | 1 | 2 | 3 | 4 | 5 | 6 | 7 | 8 | δD | δ18O |
|---|---|---|---|---|---|---|---|---|---|---|---|---|---|---|---|---|---|---|---|---|
| Dai River | S9 | Aug.2010 | | 511 | 7.2 | 22.2 | 22 | 5.5 | 80.3 | 65.2 | 107.9 | 86.3 | 72.1 | 29.9 | 3.7 | 16.7 | 0.34 | -66 | -9.4 |
| Dai River | S12 | Aug.2010 | | 485 | 7.5 | 25.8 | 18 | 7.3 | 71.3 | 41.9 | 83.7 | 83.4 | 54.3 | 27.6 | 4.0 | 15.1 | 0.30 | -69 | -9.7 |
| Dai River | S8 | Aug.2010 | | 495 | 7.3 | 22.1 | 24 | 5.8 | 68.6 | 54.9 | 96.8 | 89.3 | 62.3 | 27.2 | 4.1 | 15.9 | 0.32 | -67 | -9.6 |
| Yang River | S6 | Aug.2010 | | 507 | 7.0 | 23.3 | 17 | 4.5 | 66.0 | 51.9 | 80.5 | 107.2 | 61.6 | 32.2 | 3.8 | 16.8 | 0.30 | -65 | -9.2 |
| Yang River | S2 | Aug.2010 | | 435 | 7.3 | 7.3 | 6 | 4.9 | 63.2 | 40.2 | 92.2 | 101.2 | 59.3 | 29.9 | 4.3 | 14.0 | 0.26 | -71 | -10.1 |
| Yang River | S5 | Sep.2009 | | 718 | 8.4 | 24.4 | | | 85.2 | 12.9 | 62.0 | 107.8 | 46.1 | 52.0 | 6.4 | 17.4 | 0.36 | -45 | -5.8 |
| Yang River | S4 | Sep.2009 | | 2630 | 8.1 | 24.7 | | | 733.1 | 6.6 | 142.2 | 115.5 | 44.3 | 59.8 | 5.0 | 16.4 | 0.30 | -42 | -5.3 |
| Yang River | S6 | Sep.2009 | | 718 | 8.3 | 25.6 | | | 99.4 | 6.8 | 57.5 | 107.8 | 57.2 | 33.2 | 3.3 | 16.7 | 0.36 | -43 | -7.2 |
| Dai River | S9 | Sep.2009 | 2.0 | 560 | 8.0 | 23.6 | | | 88.8 | 6.6 | 60.5 | 92.4 | 52.1 | 98.1 | 6.0 | 23.6 | 0.55 | -46 | -5.8 |
| Dai River | S8 | Sep.2009 | | 1013 | 8.2 | 23.6 | | | 174.0 | 6.4 | 82.2 | 92.4 | 53.6 | 47.5 | 4.9 | 18.0 | 0.31 | -43 | -5.4 |
| Yang River | S6 | Jun.2008 | | 1166 | 7.5 | 12.6 | | | 81.7 | 12.6 | 71.3 | 112.5 | 60.7 | 123.2 | 10.8 | 24.2 | 0.57 | -49 | -5.5 |
| Dai River | S10 | Jun.2008 | | 1255 | 9.1 | 28.0 | | | 208.9 | 23.1 | 73.7 | 175.7 | | | | | | -44 | -3.3 |
| Dai River | S9 | Jun.2008 | | 1163 | 7.8 | 11.8 | | | 92.3 | 7.5 | 52.2 | 90.6 | 54.7 | 31.1 | 3.6 | 19.5 | 0.35 | -40 | -3.9 |

**Brackish and salt water samples**

| Water body | Sample | Date | EC | pH | T | n2 | c2 | TDS | 1 | 2 | 3 | 4 | 5 | 6 | 7 | 8 | δD | δ18O |
|---|---|---|---|---|---|---|---|---|---|---|---|---|---|---|---|---|---|---|---|
| Yang River | S3 | Sep.2009 | 34800 | 7.8 | 15.5 | 83 | 4.90 | 11289.4 | | 1684.7 | 115.5 | 251.0 | 5658.7 | 231.3 | 690.1 | 4.29 | -21 | -2.4 |
| Dai River | S12 | Sep.2009 | 47100 | 7.5 | 23.7 | | | 16766.3 | | 2416.5 | 77.0 | 412.1 | 10074.3 | 398.2 | 1306.0 | 7.64 | -12 | -1.2 |
| Dai River | S11 | Sep.2009 | 52500 | 8.5 | 23.4 | | | 1601.5 | | 258.7 | 84.7 | 79.7 | 801.4 | 32.9 | 102.7 | 0.93 | -41 | -5.2 |
| Yang River | S1 | Jun.2008 | 39800 | 8.7 | 24.2 | | | 14953.5 | | 2035.9 | 134.5 | 313.0 | 7496.0 | 270.5 | 928.3 | 5.55 | -15 | -1.1 |
| Yang River | S2 | Jun.2008 | 20200 | 8.8 | 28.5 | | 2.8 | 8328.3 | | 912.1 | 189.4 | 233.4 | 4094.2 | 147.9 | 495.9 | 3.37 | -28 | -2.5 |
| Dai River | S7 | Jun.2008 | 49500 | 8.4 | 11.5 | | | 16677.1 | 810.1 | 2261.3 | 113.9 | 349.0 | 8730.0 | 326.4 | 1084.0 | 6.36 | -11 | -0.6 |

**Seawater:**

| Sample | Date | EC | pH | T | TDS | 1 | 2 | 3 | 4 | 5 | 6 | 7 | 8 | δD | δ18O |
|---|---|---|---|---|---|---|---|---|---|---|---|---|---|---|---|
| SW1 | Aug.2010 | 45600 | 7.8 | 25.5 | 14768.3 | | 4047.0 | 148.8 | 312.7 | 8326.4 | 293.6 | 1007.0 | 5.79 | 3.8 | 1.1 |
| SW1 | Sep.2009 | 47700 | 7.8 | 24.2 | 16568.0 | | 2394.3 | 92.4 | 352.2 | 8922.0 | 322.2 | 1107.0 | 6.45 | -10 | -1.1 |
| SW2 | Sep.2009 | 39500 | 7.3 | 23.2 | 14484.8 | | 1926.6 | 107.8 | 313.2 | 7214.0 | 267.9 | 916.9 | 5.43 | -17 | -2.0 |