# Peer review of "Delineating multiple salinization processes in a coastal plain aquifer, northern China"

_Hydrology and Earth System Sciences, 2017_

## Referee Comment (RC1) · Anonymous Referee #1 · 5 Dec 2017

General comments

The specific aims of the study are reasonably clear. However, what is not clear is what new general information you hope to provide. There have been numerous studies of coastal groundwater in China (many of which are referenced). What new information will come out of this paper? There is the conceptual model, but is that different to what has been previously proposed (i.e. does this paper provide some new understanding and if so what are the current gaps in knowledge).

Additionally, how does the paper inform our understanding of coastal aquifers in general? Regional papers are useful, but to be published in International Journals they

need to convey some new understanding that ideally is applicable to other study areas. The paper commences with a review of a range of topics and seawater intrusion in a number of settings. However, it is mainly a case study and while these are important, you need to revisit those topics and explain in the conclusions the relevance to research elsewhere.

Some aspects of the study are oddly placed. Notably, the changes to groundwater levels are discussed in Sections 2.2, 2.3 and 4.1, which makes it difficult to follow. Shortening it so that only the key information is presented and adding a diagram to illustrate the trends would help immeasurably. In a similar way, Section 5 has a lot of data presentation in it as well as interpretation and much of that should be moved to Section 4.

The description of the groundwater drawdown and the geochemistry is overly long and the reader gets lost in all the details. Both these aspects could be shortened considerably and presented in a more logical order. There is a tendency to introduce concepts (eg explaining the use of Cl/Br ratios) and new data (the radioisotopes, nitrate, Br) in the discussion; Figure 10 is explained in the discussion but used much earlier. The paper needs to be reorganised. Perhaps explain in more detail how we understand geochemical processes in the introduction (e.g., the general discussion of the use of Cl/Br ratios), describe all the data in one section, and restrict section 5 to interpretations.

I agree with most of the interpretations, although for reasons explained below, the interpretation of the radioisotopes of the thermal waters (which look to come from another study) cannot be correct. I may have missed it but I don't see a clear explanation of the seasonal variations in the stable isotope values of the shallow groundwater. It must be recording seasonal recharge (?) but is that consistent with the rest of the geochemistry?

Finally, the English is difficult to read although the message is generally understandable. There are numerous places that the English needs to be corrected and I have not attempted to do this. It is not easy to write in a second language but careful proofreading of the final paper is required, which would increase its accessibility and impact.

Specific comments

Introduction

The introduction provides a comprehensive summary of the importance and threats to coastal groundwater. The threat to coastal groundwater due to sea level rise could be expanded a little as it is not just due to the interaction between seawater and the rivers, a rising sea level will induce saltwater intrusion away from the rivers and could wipe out perched freshwater bodies in some coastal aquifers.

Lines 63-70. This section does not convey much except that studies were done and the results were different. Either add a few more details about the results or just shorten it.

Lines 70-72. The definition of the hyporheic zone is older than that study, sentence needs rephrasing.

Lines 90-99. See the comments above on the aims of the study. This section needs to have a clearly articulated statement of how this review improves our general understanding.

Study Area

The geology and hydrogeology are reasonably well described needs some attention. Section 2.3 is a long and repeats some of the previous section (the drawdown and water use). It is also difficult to follow without illustrating on a Figure. Either add a map to show drawdowns or shorten this section to retain only the key facts. I am not sure that the localities (eg Zaoyuan) are on Fig. 1, which makes it difficult to follow.

Lines 122-124. Do you mean yield (abundance)

Lines 130-132. Not clear what a "complete" aquifer system is.

Lines 140-143. More detail is needed here. By how much do the water levels vary? Is it across all the area? Critically, does depletion occur near the coast? Perhaps you could show a drawdown map or a few representative hydrographs.

Lines 184-188. I presume that these are depths? What depths do the production wells pump from?

Fig. 1 would be improved by adding some hydrogeological information such as: 1) groundwater flow paths and 2) indicating the zone where seawater intrusion is observed. Is there any reason that the explanation of the colours / symbols could not go on the key rather than in the caption?

Fig. 2 uses a different set of symbols to Fig 1. Make sure that these are the same. You could also merge Fig 2 into Fig. 1 as they are related and it would be easier to get the information from these two figures if they are together.

Methods

Lines 194-201. Some more detail on the wells are needed. The Table lists a single well depth, but both here and in the table information on the screen widths are needed as characterising the geochemistry from short-screened bores is much easier than from those with longer screens.

Lines 202-213. Quote the precision for all of the parameters and lower detection limits where important.

Lines 221-238. This is a standard technique and the description of it could be shortened.

Results

Section 4.1 repeats some of the historical information that is section 2 (the description of the cones of depression). Again, it is done without any illustration. As this is really background material, it probably is better to merge it into Section 2 which would avoid

some of the long descriptions. It is difficult to understand the water level changes due to pumping as they are written. Given that the paper is long and is mainly focussed on geochemistry, it would be worth presenting the changes to groundwater levels in a single section (currently there is information in Sections 2.2, 2.3 and 4.1), shortening it so that only the key information is presented, and adding a diagram.

Section 4.2.

Lines 217-281. I am not sure that you have enough data to discern a seasonal variation. Is the variation the same as in the local rainfall (I presume that there are data)? If so you could just note that the rivers have stable isotope trends that follow those of the rainfall and shorten the detail in this section.

Lines 294-302. Suggest presenting the rainfall data first as it is part of the general background data against which you can compare your observations.

Line 302. Not clear what you mean by this.

Section 4.3 & Section 5.1

Section 4.3 does a good description of the major ion geochemistry. However, there is much basic description in section 5.1 (e.g., lines 386-420). Also there are data introduced in Section 5 (nitrate and Br) that do not appear in Section 4.

Overall splitting the data up in this way makes the paper long and convoluted. You need to decide where information goes and be consistent. Presenting all of the descriptive material (including water types) in section 4 and restricting interpretations of the data to Section 5 would make more sense. In which case the descriptive material on lines 386-420 could be merged into section 4.3.

The amount of description is also long. It is good to present the data in the text and I get very frustrated with papers that just refer to data in tables or figures without discussion, but there is a lot of detail presented in this study. Both for the stable isotope and the major ion data I would suggest cutting the detail down and presenting what you

think is necessary. For example, do you need to describe the water types or would an explanation of the variations in salinity and general water chemistry be sufficient for this study? The Piper diagram does not show the types in any case but the separation of the waters is clear.

You also have a facies diagram (Fig. 8) and bivariate plots (Fig. 10). I am not convinced that you need both as surely the processes of freshening and intrusion could be shown on the bivariate plots?

Discussion

Lines 362-368. If the thermal water has measurable tritium then it has some component with a mean residence time of less than ~100 years. Given that it has old 14C "ages" it looks to be a mix of old water (zero tritium, low 14C) and young water (high tritium, high 14C). In which case the 14C ages are meaningless. The mixing will also affect the stable isotope ratios and the interpretation of palaeowaters (although the older component may still have a past climate signal). I am not convinced that the thermal waters are important to this story, but if they are going to be included, then they need to be interpreted correctly.

Lines 386-405. See above, this is description and belongs earlier.

Lines 406-409. Again, you are introducing new data. Collect all the descriptions of the data into Section 4.

Lines 441-458. This section describes some of the consequences of salinization, which would be better discussed towards the end of the paper after you have discussed the processes.

Lines 459-470. It is confusing to only introduce Fig. 10 here as most of its use is to describe water types, the presentation of which was much earlier in the paper. In any case, I am not convinced that the details on the water types adds much of substance to the paper. It lengthens the text and the salinization and freshening trends can be

illustrated on the other diagrams.

Section 5.2.2. See comments above. The interpretation of the radioisotopes cannot be correct. In addition, the data is being re-presented here (lines 513-515).

Lines 530-544. There is a fair amount of introductory explanation in this section, which could have been presented earlier. More importantly, more new data (the Br and the Cl/Br ratios) is being introduced. It make the paper very difficult to follow when data is described piecemeal rather than in one section.

Section 5.3

The conceptual model is reasonable, but is it new? In this section or perhaps in Section 6, you should outline more clearly how your study has improved the understanding of seawater intrusion in this region (or how your model compares with conventional wisdom) and also how it fits in with current global understanding. The paper commenced by discussing a range of global issues and summarising a number of key studies, but it is not clear as to how the paper informs that global research and what relevance it may have to researchers working elsewhere.

---

## Referee Comment (RC2) · Anonymous Referee #2 · 21 Dec 2017

General comment

The paper by Han describes the groundwater salinization processes and its inducement of a coastal aquifer. In the paper, a database of chemical and isotopic data is discussed to evaluate the hydrogeochemical processes governing groundwater flow in the aquifer and the overall quality of the water resources. The paper is mainly descriptive, and applies some geochemical processes to explain the behavior of the aquifer. Nevertheless, the way it has been focused is constrained to a regional study. The introduction is overelaborate without giving a straightforward idea of what the paper would like to present. The way how the geochemical data is explained is pretty much like

those papers that are cited in the references. Meanwhile, it lacks new understandings of coastal auifers in general. The paper is organized in some way but for a publication in a top Internation Journal, I consider that it does not constitute a valuable scientific contribution.

---

## Author Comment (AC1) · 12 Feb 2018

**Responses to Editor and Reviewers**

Dr. Dongmei HAN

Key Laboratory of Water Cycle & Related Land Surface Processes

Institute of Geographic Sciences and Natural Resources Research

Chinese Academy of Sciences

Beijing 100101, P.R. China

Fax: +86-10-64889849

Tel: +86-10-64889367

Beijing, 12 February, 2018

To:

Prof. Hu Bill

Associate Editor

Hydrology and Earth System Sciences

Dear Prof. Hu,

We are submitting the revised manuscript titled "Delineating multiple salinization processes in a coastal plain aquifer, northern China: hydrochemical and isotopic evidence" (HESS -2017-617) to **Hydrology and Earth System Sciences**. Following the constructive comments from two reviewers, the authors have completed revisions on the previous manuscript, addressing each point raised by the reviewers. We gratefully acknowledge their generous contribution and feedback.

**Reply to the anonymous Referee #1:**
We would like to thank you for the very valuable comments on our manuscript, we think these comments will help to improve its quality greatly. We have attempted to address each of the comments point-by-point:

Anonymous Referee #1:

General comments

The specific aims of the study are reasonably clear. However, what is not clear is what new general information you hope to provide. There have been numerous studies of coastal groundwater in China (many of which are referenced). What new information will come out of this paper? There is the conceptual model, but is that different to what has been previously proposed (i.e. does this paper provide some new understanding and if so what are the current gaps in knowledge). Additionally, how does the paper inform our understanding of coastal aquifers in general? Regional papers are useful, but to be published in International Journals they need to convey some new understanding that ideally is applicable to other study areas. The paper commences with a review of a range of topics and seawater intrusion in a number

**Response:** Agree, changes made. We have revised the objectives of this paper as outlined in Lines 104-120, as well as highlighting the relationship between previous work in the study area, and the novel contribution made in this case (building on past work).

In China, there are 18,000 kilometers of coastline. Approximately 12% of the population (>100 million people) are distributed in these low elevation coastal zones, which are highly vulnerable to water supply stress caused by seawater intrusion. The most serious seawater intrusion (SWI) in China occurs in the Circum-Bohai-Sea region, with the estimated area affected being 2457 $km^2$, increasing at a rate of 6 $km^2$/a since the 1980s. This study comprehensively delineates the interaction between marine surface water and groundwater as well as mixing between low-temperature fresh groundwater and thermal waters in Quaternary aquifers of the Circum-Bohai coastal plain. Few previous studies have examined cases involving multiple hydrochemical end-members and salinization sources, including both marine water and deep geothermal water. The study thus provides a novel case study, using a series of geochemical/isotope indicators to separate the influence of these different mechanisms. The additional aspect delineating anthropogenic pollution as a distinct process contributing to salinization is also relatively rare in the literature.

The salinization processes occurring in coastal area around the Circum-Bohai Sea-Region examined in this study are quite different from coastal carbonate aquifers in the Dalian area (e.g., Han et al., 2015) and Quaternary aquifers with paleo-seawater residue (brines) Laizhou Bay (e.g., Han et al., 2014). The study is therefore both novel, in that the contribution of thermal water up-welling is delineated in addition to SWI influence and of great practical significance, as similar processes may be occurring elsewhere in the Circum-Bohai or other regions.

*Lines 109-120:* "In the past 30 years, many studies have investigated seawater intrusion and its influencing factors in the region using hydrochemical analysis (Xu, 1986; Yang et al., 1994, 2008; Chen and Ma, 2002; Sun and Yang, 2007; Zhang, 2012) and numerical simulations (Han, 1990; Bao, 2005; Zuo, 2009). However, these studies have yet to provide clear resolution of the different mechanisms contributing to salinization, and have typically ignored the role of anthropogenic pollution and groundwater-surface water interaction. This study is thus a continuation of previous investigations of the region, using a range of hydrochemical and stable isotopic data to delineate the major processes responsible for increasing groundwater salinity, including lateral sub-surface sea-water intrusion, vertical leakage of marine-influenced surface water, induced mixing of saline geothermal water, and anthropogenic pollution. The goal is to obtain a more robust conceptual model of the interconnections between the various water sources under the impact of groundwater exploitation. The results provide significant new information to assist water resources management in the coastal plain of Bohai Bay, and other similar coastal areas globally."

discussed in Sections 2.2, 2.3 and 4.1, which makes it difficult to follow. Shortening it so that only the key information is presented and adding a diagram to illustrate the trends would help immeasurably. In a similar way, Section 5 has a lot of data presentation in it as well as interpretation and much of that should be moved to Section 4.

**Response:** We agree and have made substantial changes in response to this comment, and related subsequent comments (below), including a re-organisation of the paper to reduce repetition and give it a logical flow between sections. Information relating to particular topics (such as water level data, historic monitoring of saline intrusion etc.) has been removed from the results section (4.1) and is now consolidated in a revised introductory section (2.3; Lines 170-219). As suggested, we have also included a new diagram (Supplementary Fig. S1) to show the distribution of groundwater levels in 1986, 1998, 2004, and 2010:

[Figure]

Fig. S1 Maps showing the distribution of groundwater level contours in shallow aquifers (a) in 1986 (from Han, 1988), (b) in 1998 (from Zuo, 2006), (c) in 2004 (from Zuo, 2006), and (d) in 2010 (this study)

Additionally, we significantly reorganized the Discussion (section 5). Data previously included within the Discussion (e.g. in section 5.2.1) has now been consolidated in the introduction (section 2.3) and/or Results (4). Information in the previous version relating to hydrochemical 'types' was deemed unnecessary to define the key end-members and derive mixing trends (as suggested by the reviewer), and was thus removed.

The description of the groundwater drawdown and the geochemistry is overly long and the reader gets lost in all the details. Both these aspects could be shortened considerably and presented in a more logical order. There is a tendency to introduce concepts (eg explaining the use of Cl/Br ratios) and new data (the radioisotopes, nitrate, Br) in the discussion; Figure 10 is explained in the discussion but used much earlier. The paper needs to be reorganised. Perhaps explain in more detail how we understand geochemical processes in the introduction (e.g., the general discussion of the use of Cl/Br ratios), describe all the data in one section, and restrict section 5 to interpretations.

**Response:** Agree, changes made. As discussed above, a full-reorganisation of the paper has been conducted in response to the reviewer's recommendations. Much of the introductory material has been significantly condensed and consolidated. We provide background about use of various geochemical tracers (including ionic ratios) in studies of groundwater salinization in the introduction (lines 88 to 103) and have condensed the description of the data into two major sections – stable isotopes (section 4.1) and Water salinity/Dissolved ions (section 4.2). The order of all figures (including Figure 10) has been revised such that these are introduced in proper order, in conjunction with the relevant material discussed in the text, and only those relevant to the major findings are included (e.g., Figures 1&2 have merged; Figures 9, 11, 12 and 14 were removed, Figures 10 and 13 were moved to earlier in the manuscript where they are relevant (now Figures 6 and 7).

I agree with most of the interpretations, although for reasons explained below, the interpretation of the radioisotopes of the thermal waters (which look to come from another study) cannot be correct. I may have missed it but I don't see a clear explanation of the seasonal variations in the stable isotope values of the shallow groundwater. It must be recording seasonal recharge (?) but is that consistent with the rest of the geochemistry?

**Response:** Agree, changes made. We re-examined the radioisotope data, and decided to remove this from the manuscript, as these were collected by another group some time ago, and the data quality can't be verified. We also looked closely at the stable isotopic data, in particular seasonal variations in these in some of the wells. This is noted in the revised section 4.1 on stable isotopes (Lines 278-283):

"Slight seasonal variation was evident in the groundwater isotope compositions; shallow groundwater from the dry season (n = 12) showed $\delta^{18}O$ and $\delta^{2}H$ values from -7.2 to -4.2‰ (mean = -5.7‰) and $\delta^{2}H$ values from -56 to -39‰ (mean = -48‰); while during the wet season (n = 31) $\delta^{18}O$ and $\delta^{2}H$ values ranged from -11.0 ~ -5.3‰ (mean = -6.9‰) and -76 ~ -43‰ (mean = -51‰), respectively. Some variability was also evident in deep groundwater compositions, although only three deep samples were collected during the dry season."

Some of the observed variability (in shallow groundwater) is indeed interpreted as reflecting seasonal recharge (Lines 363-372):

"The two fresh end-members were selected to represent a range of different groundwater compositions/recharge sources, from shallow water that is impacted by infiltration of partially

evaporated recharge (fresh but with enriched $\delta^{18}$O) to deeper groundwater unaffected by such enrichment (fresh and with relatively depleted $\delta^{18}$O). The narrower range and relatively enriched stable isotopes in shallow groundwater samples collected during the dry season compared with the wet season indicate some influence of seasonal recharge by either rainfall (fresh, with relatively depleted stable isotopes) or irrigation water subject to evaporative enrichment (more saline, with enriched stable isotopes and high nitrate concentrations; Currell et al., 2010) and/or surface water leakage. While there is overlap in the isotopic and hydrochemical compositions of shallow and deep groundwater (Fig. 3 & Fig. 4), this effect appears to only affect the shallow aquifer."

Finally, the English is difficult to read although the message is generally understandable. There are numerous places that the English needs to be corrected and I have not attempted to do this. It is not easy to write in a second language but careful proofreading of the final paper is required, which would increase its accessibility and impact.

**Response:** The manuscript has been thoroughly revised and polished carefully for readability and English language according to this recommendation.

Specific comments
Introduction
The introduction provides a comprehensive summary of the importance and threats to coastal groundwater. The threat to coastal groundwater due to sea level rise could be expanded a little as it is not just due to the interaction between seawater and the rivers, a rising sea level will induce saltwater intrusion away from the rivers and could wipe out perched freshwater bodies in some coastal aquifers.

**Response:** Partly agree, change made. Undoubtedly, sea level rise is one factor threatening coastal groundwater, and this is now mentioned in the introduction (see line 60). However, we are of the view (following that of Ferguson and Gleeson, 2012) that the effects of sea-level rise on sub-surface saline intrusion are likely to be a relatively minor in comparison to ingress/inundation of tidal water into surface estuaries and effects related to increasing groundwater extraction in response to climate stress. In addition, because we are focusing on characterization of current and historical salinization processes, and are not conducting any modeling of future scenarios, we believe further in-depth discussion of sea level rise is not warranted (this is covered by other recent studies cited in the introduction, such as Werner et al., 2013).

Lines 63-70. This section does not convey much except that studies were done and the results were different. Either add a few more details about the results or just shorten it. Lines 70-72. The definition of the hyporheic zone is older than that study, sentence needs rephrasing.

**Response:** Agree. Most of this text was removed and the introduction condensed to focus on background issues relevant to the topic (e.g. the discussion of processes in the Nile Delta, and other modeling studies are not directly related to our study). See revised version of lines 62 to 74.

Lines 90-99. See the comments above on the aims of the study. This section needs to have a clearly articulated statement of how this review improves our general understanding.

**Response:** Agree, changes made. We have rewritten the objectives of this study, as follows (Lines 104-120):

"This study examines the Yang-Dai River coastal plain in Qinhuangdao City, Hebei province, north China, specifically focusing on salinization of fresh groundwater caused by groundwater exploitation in Zaoyuan well field and surrounding areas. The study investigates groundwater salinization processes and interactions among surface water, seawater and geothermal groundwater in a dynamic environment, with significant pressure on water resources. Qinhuangdao is an important port and tourist city of northern China. In the past 30 years, many studies have investigated seawater intrusion and its influencing factors in the region using hydrochemical analysis (Xu, 1986; Yang, 1994, 2008; Chen and Ma, 2002; Sun and Yang, 2007; Zhang, 2012) and numerical simulations (Han, 1990; Bao, 2005; Zuo, 2009). However, these studies have yet to provide clear resolution of the different mechanisms contributing to salinization, and have typically ignored the role of anthropogenic pollution and groundwater-surface water interaction. This study is thus a continuation of previous investigations of the region, using a range of hydrochemical and stable isotopic data to delineate the major processes responsible for increasing groundwater salinity, including lateral sub-surface sea-water intrusion, vertical leakage of marine-influenced surface water, induced mixing of saline geothermal water, and anthropogenic pollution. The goal is to obtain a more robust conceptual model of the interconnections between the various water sources under the impact of groundwater exploitation. The results provide significant new information to assist water resources management in the coastal plain of Bohai Bay, and other similar areas in China and globally."

Study Area
The geology and hydrogeology are reasonably well described needs some attention. Section 2.3 is a long and repeats some of the previous section (the drawdown and water use). It is also difficult to follow without illustrating on a Figure. Either add a map to show drawdowns or shorten this section to retain only the key facts. I am not sure that the localities (eg Zaoyuan) are on Fig. 1, which makes it difficult to follow.

**Response:** Agree, changes made. The structure of the manuscript has been reorganized, as discussed above. Information about groundwater usage and drawdown has been significantly condensed into a concise new section – 2.3 Groundwater usage and seawater intrusion history (lines 170-219). The location of the Zaoyuan well field is now clearly shown using the legend of the revised Fig. 1. Water level maps have now been produced to show changes in these patterns through time (Figure S1).

Lines 122-124. Do you mean yield (abundance)

**Response:** Agree, change made. We re-wrote this sentence for clarity (Lines 143-145): 'Groundwater in the area includes water in the Quaternary porous sediment as well as fractured bedrock in the northern platform area. Fractured rock groundwater volume mainly

depends on the degree of weathering and the nature and regularity of fault zones (Fig. 1).'

Lines 130-132. Not clear what a "complete" aquifer system is.

**Response:** Agree, change made. The text in question was deleted in the revised description of the aquifer system (Lines 152-159).

Lines 140-143. More detail is needed here. By how much do the water levels vary? Is it across all the area? Critically, does depletion occur near the coast? Perhaps you could show a drawdown map or a few representative hydrographs.

**Response:** Agree, changes made. As discussed above, this information is now included in a new section (2.3) along with a new figure showing drawdown patterns during four time periods (Figure S1). Figure 2 also shows representative changes in groundwater levels through time (continuous monitoring series) in three shallow monitoring wells, locations of which are shown on the revised Fig. 1. This information gives a clear picture of where and by how much water levels vary in the region with respect to groundwater usage.

Lines 184-188. I presume that these are depths? What depths do the production wells pump from?

**Response:** Agree, changes made. We checked the previous investigation materials and confirmed that the depths for the production wells in Zaoyuan well field are approximately 15 -20m, as noted in the revised text (line 184).

Fig. 1 would be improved by adding some hydrogeological information such as: 1) groundwater flow paths and 2) indicating the zone where seawater intrusion is observed. Is there any reason that the explanation of the colours/ symbols could not go on the key rather than in the caption?

**Response:** Agree, changes made. We added arrows showing the groundwater flow direction and major seawater intrusion zones – defined as areas with >250 mg/L of chloride (see revised Fig. 1).

Fig. 2 uses a different set of symbols to Fig 1. Make sure that these are the same. You could also merge Fig 2 into Fig. 1 as they are related and it would be easier to get the information from these two figures if they are together.

**Response:** Agree, changes made. In accordance with the recommendation we have merged Fig. 2 into Fig. 1, and all relevant legend symbols are now included on the same key.

Methods
Lines 194-201. Some more detail on the wells are needed. The Table lists a single well depth, but both here and in the table information on the screen widths are needed as characterising the geochemistry from short-screened bores is much easier than from those with longer screens.

**Response:** Agree, changes made. We agree that understanding the well construction details, including screened interval length, is important when interpreting groundwater geochemistry data. Due to an absence of monitoring wells in the study, we unfortunately had no choice but to utilize production wells for our sampling. The well depths provided in Table 1 represent total well depths and in most cases the screened interval spans 5 to 15m above this depth (See new text lines 225-227).

Lines 202-213. Quote the precision for all of the parameters and lower detection limits where important.

**Response:** Agree, this information has now been supplied in the methods section (Lines 243-247).

Lines 221-238. This is a standard technique and the description of it could be shortened.

**Response:** Agree, changes made. While the ionic delta values and saturation indices are potentially interesting, but now believe that (as the reviewer suggests below) the major processes of interest can be determined without looking at these indicators. Hence these methods and results were removed, to make the paper more concise and focused.

Results
Section 4.1 repeats some of the historical information that is section 2 (the description of the cones of depression). Again, it is done without any illustration. As this is really background material, it probably is better to merge it into Section 2 which would avoid some of the long descriptions. It is difficult to understand the water level changes due to pumping as they are written. Given that the paper is long and is mainly focused on geochemistry, it would be worth presenting the changes to groundwater levels in a single section (currently there is information in Sections 2.2, 2.3 and 4.1), shortening it so that only the key information is presented, and adding a diagram.

**Response:** Agree, changes made. As discussed above, we have moved all of the information about groundwater dynamics into the background section (2.3 Groundwater usage and seawater intrusion history), updated Figure 1 and included a new figure showing spatial changes in water levels (Fig S1). In addition, we also include a diagram (Fig. S2, below) showing the seawater intrusion history (shown as chloride concentrations and SWI area) along with historic rainfall variation, indicated by cumulative fluctuations in the monthly average rainfall, and added the following text (Lines 205-208):

"As indicated in Figure S2, the severity of seawater intrusion (indicated by changes in Cl concentration, and the total area impacted by SWI, as defined by the 250mg/L Cl contour) correlates with periods of below average rainfall – indicated by monthly cumulative rainfall departure (CRD, Weber and Stewart, 2004)."

[Figure]

Fig. S2 Graph showing the temporal variation of the monthly cumulative rainfall departure (CRD, Weber and Stewart, 2004), monthly precipitation, the average concentration of the chloride ion in groundwater of the study area (dark blue) and surface area with >250 mg Cl/L (yellow) between 1963 and 2008 (data from Zang et al., 2010).

Section 4.2.
Lines 217-281. I am not sure that you have enough data to discern a seasonal variation. Is the variation the same as in the local rainfall (I presume that there are data)? If so you could just note that the rivers have stable isotope trends that follow those of the rainfall and shorten the detail in this section.

**Response:** Agree, Changes made. This information has been revised into a dedicated section on the water stable isotopes (4.1). We include a new diagram (Fig. S3, below) showing that rivers have stable isotope trends that generally follow those of the rainfall, and have noted this in the text (Lines 267 to 273):

"Stable isotope compositions for surface water appear to exhibit significant seasonal variation (Fig. S3); for Yang River samples from the relatively dry season (June 2008, n = 3) had mean $\delta^{18}O$ and $\delta^{2}H$ values of -3.0‰ -31‰, respectively; samples from the wet season (August 2009 and September 2010, n = 6) had mean $\delta^{18}O$ and $\delta^{2}H$ values of -6.6‰ -48‰, respectively. Dai River samples showed similar results; the dry season mean $\delta^{18}O$ and $\delta^{2}H$ values (n = 3) were -2.6‰ and -32‰, respectively; wet season samples (n = 7), had mean $\delta^{18}O$ and $\delta^{2}H$ values of -6.6‰ and -49‰, respectively (Fig. 3)."

[Figure]

Fig. S3 Graph showing $\delta^2H$ vs. $\delta^{18}O$ of water samples in rainfall and river water. Dry season- July to October; wet season- November to June.

The description of the changes in isotopic values in groundwater sampled in different seasons has been updated. We note that there is some variability in shallow groundwater indicating an influence from differences in seasonal recharge mechanism and/or amount (Lines 278-283):

"Slight seasonal variation was evident in the groundwater isotope compositions; shallow groundwater from the dry season (n = 12) showed $\delta^{18}O$ and $\delta^2H$ values from -7.2 to -4.2‰ (mean = -5.7‰) and $\delta^2H$ values from -56 to -39‰ (mean = -48‰); while during the wet season (n = 31) $\delta^{18}O$ and $\delta^2H$ values ranged from -11.0 ~ -5.3‰ (mean = -6.9‰) and -76 ~ -43‰ (mean = -51‰), respectively. Some variability was also evident in deep groundwater compositions, although only three deep samples were collected during the dry season."

Lines 294-302. Suggest presenting the rainfall data first as it is part of the general background data against which you can compare your observations.

**Response:** Agree, change made. The section describing the rainfall isotopes has been moved to the beginning of section 4.1 (lines 262 to 265):
"The local meteoric water line (LMWL, $\delta^2H$=6.6 $\delta^{18}O$+0.3, n=64, $r^2$=0.88) is based on $\delta^2H$ and $\delta^{18}O$ mean monthly rainfall values between 1985 and 2003 from Tianjin station some 120 km SW of Qinhuangdao City (IAEA/WMO, 2006). Due to similar climate and position relative to the coast, this can be regarded as representative of the study area."

Line 302. Not clear what you mean by this.

**Response:** Agree, change made. We rephrased this sentence so the meaning is clearer (Line 288 to 291: "The local seawater plots below (more negative) than typically assumed values (e.g. VSMOW = 0‰) for both $\delta^2H$ and $\delta^{18}O$, and this water appears to represent an end-member involved in mixing with meteoric-derived waters in both ground and surface water (Fig. 3)."

Section 4.3 & Section 5.1

Section 4.3 does a good description of the major ion geochemistry. However, there is much basic description in section 5.1 (e.g., lines 386-420). Also there are data introduced in Section 5 (nitrate and Br) that do not appear in Section 4.

**Response:** Agree, changes made. We have rewritten and condensed the results and discussion section, so that the major ion geochemistry data is contained in the results (now section 4.2), while much of the new data introduced and discussed in the discussion (section 5.1) has been condensed or removed. The nitrate data were moved into the results section (Line 300), while we have decided not to discuss the Br data, as these were not necessary for explaining the observed salinization processes.

Overall splitting the data up in this way makes the paper long and convoluted. You need to decide where information goes and be consistent. Presenting all of the descriptive material (including water types) in section 4 and restricting interpretations of the data to Section 5 would make more sense. In which case the descriptive material on lines 386-420 could be merged into section 4.3.

**Response:** Agree. The revised manuscript ensures that description and reporting of data is confined to the results section (4), while interpretations are given in the discussion (section 5). The only exceptions area where data were derived from other studies (see below) in which case they are not reported as our results in section 4, but are rather introduced in the discussion of interpretations as important/complementary evidence.

The amount of description is also long. It is good to present the data in the text and I get very frustrated with papers that just refer to data in tables or figures without discussion, but there is a lot of detail presented in this study. Both for the stable isotope and the major ion data I would suggest cutting the detail down and presenting what you think is necessary. For example, do you need to describe the water types or would an explanation of the variations in salinity and general water chemistry be sufficient for this study? The Piper diagram does not show the types in any case but the separation of the waters is clear.

**Response:** Agree, changes made. Some sections in the discussion have been completely removed and/or shortened - e.g. sections 5.2.1 and 5.2.2 have now been integrated into a more concise discussion of hydrochemical indicators of mixing processes (Section 5.1). We have retained some basic discussion of water types and their relationship with different salinity sources (e.g. Lines 394 to 402), however this is significantly shorter than in the original manuscript. Discussion of saturation indices and ionic delta values has also been removed from the manuscript, as most of the trends in hydrochemical evolution can be described without reference to these techniques (see revised section 5.3). Overall the discussion has reduced in length from 257 to 172 lines.

You also have a facies diagram (Fig. 8) and bivariate plots (Fig. 10). I am not convinced that you need both as surely the processes of freshening and intrusion could be shown on the bivariate plots?

**Response:** Disagree. In the revised version, we have retained both figures, as we think they

are both important in explaining the role of thermal groundwater mixing (e.g. Na/Cl and $Ca/SO_4 +HCO_3$) as well as freshening and intrusion during seawater intrusion. We believe it is not possible to represent all of the necessary trends related to these processes on each of the diagrams alone, and there is value in inclusion of both (particularly with respect to demonstrating the role of base exchange).

Discussion
Lines 362-368. If the thermal water has measurable tritium then it has some component with a mean residence time of less than ~ 100 years. Given that it has old $^{14}$C "ages" it looks to be a mix of old water (zero tritium, low $^{14}$C) and young water (high tritium, high $^{14}$C). In which case the $^{14}$C ages are meaningless. The mixing will also affect the stable isotope ratios and the interpretation of palaeowaters (although the older component may still have a past climate signal). I am not convinced that the thermal waters are important to this story, but if they are going to be included, then they need to be interpreted correctly.

**Response:** Agree, changes made. After re-examination of the radioisotope data in the original source document, we decided that the data cannot be verified, and as such we have removed them from the manuscript. The associated discussion of $^{14}$C and $^{3}$H data has been removed from the manuscript. We believe the stable isotopes, major and minor ions – particularly strontium - give sufficient insight into salinization processes without the need for these data. A new section discussing the strontium data, including Sr/Cl ratios has now been included which we believe provides the clearest evidence of thermal/low temperature water mixing (lines 403 to 418):

"Stronger evidence of mixing of the geothermal water in the Quaternary aquifers (particularly deep groundwater) is provided by examining strontium concentrations in conjunction with chloride (Fig. 8). The geothermal water from Danihe geothermal field has much higher Sr concentrations (up to 89.8 mg/L) than seawater (5.4-6.5 mg/L in this study), due to Sr-bearing minerals (i.e., celestite, strontianite) with Sr contents of 300-2000 mg/kg present in the bedrock (Hebei Geology Survey, 1987). Groundwater sampled from near the geothermal field in this study has the highest Sr concentrations e.g., G9 with Sr concentrations ranging from 7.4 to 11.6 mg/L, and G19 from 4.9 to 7.1 mg/L.

The plot of chloride versus strontium concentrations (Fig. 8) shows that these samples and others (e.g., G16, G20, G27, G29) plot close to a mixing line between fresh low-temperature and saline thermal-groundwater. Mass ratios of Sr/Cl in these samples are also elevated relative to seawater by an order of magnitude or more (e.g. Sr/Cl >$5.0 \times 10^{-3}$, compared to $3.9 \times 10^{-4}$ in seawater). Other samples from closer to the coast (e.g. G4) also approach the thermal-low temperature mixing line, indicating probable input of thermal water. Samples collected from the Zaoyuan well field generally plot closer to the Sr/Cl seawater mixing line (consistent with salinization largely due to marine water – Fig. 8); however, samples mostly plot slightly above the mixing line with additional Sr, which may indicate more widespread (but volumetrically minor) mixing with the thermal water in addition to seawater."

Lines 386-405. See above, this is description and belongs earlier.

**Response:** Agree, changes made. This descriptive information about salinization trends in response to pumping has all been consolidated into section 2.3 of the manuscript in background information (Groundwater usage and seawater intrusion history).

Lines 406-409. Again, you are introducing new data. Collect all the descriptions of the data into Section 4.

**Response:** Disagree, although changes made. The nitrate data we are reporting here (in surface water from Bohai Bay) was not collected in our study but reported in other sources, so we did not describe this in the results section as suggested. Our nitrate concentration data collected for this study is reported in section 4 (lines 325-327). For improved readability and structure, we have consolidated and condensed the discussion of nitrate data in a new section of the discussion (5.2 Anthropogenic pollution of groundwater), which discusses our collected nitrate data in the context of the Bohai bay surface water data published elsewhere, and compares the ratios between the two.

Lines 441-458. This section describes some of the consequences of salinization, which would be better discussed towards the end of the paper after you have discussed the processes.

**Response:** Agree, changes made. In the revised version we moved this and other information regarding consequences of salinization to a new section at the end of the discussion; '5.4 Conceptual model of salinization and management implications'. See Lines 460-501.

Lines 459-470. It is confusing to only introduce Fig. 10 here as most of its use is to describe water types, the presentation of which was much earlier in the paper. In any case, I am not convinced that the details on the water types adds much of substance to the paper. It lengthens the text and the salinization and freshening trends can be illustrated on the other diagrams.

**Response:** Partly Agree, changes made. As described above, the discussion of water types has been significantly condensed in section 5.1. However, we still feel there is value in including the hydrochemical facies diagram to discuss specifically the changes in major ion composition due to salinization/freshening, with reference to Figure 11. This is now included in section 5.3 under 'Hydrochemical evolution during salinization'.

Section 5.2.2. See comments above. The interpretation of the radioisotopes cannot be correct. In addition, the data is being re-presented here (lines 513-515).

**Response:** Agree. We deleted the text related to the interpretation of the radio isotopes.

Lines 530-544. There is a fair amount of introductory explanation in this section, which could have been presented earlier. More importantly, more new data (the Br and the Cl/Br ratios) is being introduced. It makes the paper very difficult to follow when data is described piecemeal rather than in one section.

**Response:** Agree, changes made. The background information regarding ionic ratios was

moved back into section 1. We decided that the Br and Cl/Br ratios were ultimately not required to explain the salinization processes and did not add much additional insight into the processes, therefore these data and the text referred to has been removed.

The conceptual model is reasonable, but is it new? In this section or perhaps in Section 6, you should outline more clearly how your study has improved the understanding of seawater intrusion in this region (or how your model compares with conventional wisdom) and also how it fits in with current global understanding. The paper commenced by discussing a range of global issues and summarising a number of key studies, but it is not clear as to how the paper informs that global research and what relevance it may have to researchers working elsewhere.

**Response:** Agree, changes made. In the revised version of the manuscript we have tried to be much more clear about the new contribution to understanding salinization processes our study has made, as highlighted in the introduction section (lines 109 to 120). This study for the first time clearly delineates between multiple competing processes responsible for salinization, namely, marine water intrusion (by sub-surface and surface pathways), mixing with saline thermal water due to intensive extraction, and anthropogenic pollution. We have highlighted the value of the updated conceptual model in section 5.3 (Lines 473-489):

"A conceptual model of the groundwater flow system in the Yang-Dai River coastal plain is summarized in Fig. 11. This model presents an advance on the previous understanding of the study area, by delineating four major processes responsible for groundwater salinization in this area. These are: 1. Seawater intrusion by lateral sub-surface flow; 2. Interaction between saline surface water and groundwater (e.g. vertical leakage of saline water from the river estuaries); 3. Mixing between low-temperature groundwater and deep geothermal water; and 4. Irrigation return-flow and associated anthropogenic contamination. Both the lateral and vertical intrusion of saline water are driven by the long-term over-pumping of groundwater from fresh aquifers in the region. The irrigation return-flow from local agriculture results from over-irrigation of crops, and is responsible for extensive nitrate pollution (up to 340 mg/L $NO_3^-$ in groundwater of this area) probably due to dissolution of fertilizers during infiltration. The somewhat enriched stable isotopes in shallow groundwater (more pronounced in the dry season) also indicate that such return-flow may recharge water impacted by evaporative salinization into the aquifer. The geothermal water, with distinctive chemical composition (e.g. depleted stable isotopes, high TDS, Ca and Sr concentrations), is also demonstrated in this study to be a significant contributor to groundwater salinization, via upward mixing. The study area is therefore in a situation of unusual vulnerability, in the sense that it faces salinization threats simultaneously from lateral, downward and upward migration of saline water bodies."

Additionally, we have attempted to draw out some of the key management implications for this area and other similar areas globally, to increase the global relevance of the paper (Lines 490-501):

"According to drinking water standards and guidelines from China Environmental Protection Authority (GB 5749-2006) and/or US EPA and WHO, chloride concentration in drinking water should not exceed 250 mg/L. At the salinity levels observed in this study - many samples impacted by salinization contain >500mg/L of chloride (Table 1) - a large amount of groundwater is now or will soon be unsuitable for domestic usage, as well as irrigation or industrial utilization. So far, this has enhanced the scarcity of fresh water resources in this region, leading to a cycle of groundwater level decline → seawater intrusion → loss of available freshwater → increased pumping of remaining freshwater. If this cycle continues, it is likely to further degrade groundwater quality and restrict its usage in the future. Such a situation is typical of the coastal water resources 'squeeze' highlighted by Michael et al., (2017). Alternative management strategies, such as restricting water usage in particular high-use sectors, such as agriculture, industry or tourism, that are based on a comprehensive assessment of the social, economic and environmental benefits and costs of these activities, warrants urgent and careful consideration."

Anonymous Referee #2:
General comment
The paper by Han describes the groundwater salinization processes and its inducement of a coastal aquifer. In the paper, a database of chemical and isotopic data is discussed to evaluate the hydrogeochemical processes governing groundwater flow in the aquifer and the overall quality of the water resources. The paper is mainly descriptive, and applies some geochemical processes to explain the behavior of the aquifer. Nevertheless, the way it has been focused is constrained to a regional study. The introduction is overelaborated without giving a straightforward idea of what the paper would like to present. The way how the geochemical data is explained is pretty much like those papers that are cited in the references. Meanwhile, it lacks new understandings of coastal auifers in general. The paper is organized in some way but for a publication in a top Internation JournaI, I consider that it does not constitute a valuable scientific contribution.

**Response:** Partly agree, changes made. Significant changes have been made to the revised version of the manuscript (see response to previous reviewer's comments), which we believe addresses the issues raised here by Reviewer #2. For example, the introduction has been condensed, but now highlights the rationale and scope of this study and the new contribution it makes to understanding salinization processes both in the study area, and more broadly (see lines 104 to 120, and section 5.3 of the revised paper.

---

## Author Comment (AC2) · 12 Feb 2018

Please see the attached file (Revision notes).

Please also note the supplement to this comment:
https://www.hydrol-earth-syst-sci-discuss.net/hess-2017-617/hess-2017-617-AC2-supplement.pdf

---

## Author Comment (AC3) · 12 Feb 2018

[revised manuscript text omitted]
| ***Deep Groundwater samples:*** | | | | | | | | | | | | | | | | | | | | | |
| Fresh Groundwater | G25 | Aug.2010 | 4 | 95 | | 444 | 6.6 | 23.5 | 41 | 4.4 | 68.1 | 71.0 | 48.2 | 89.3 | 52.5 | 35.9 | 1.1 | 10.4 | 0.25 | -56 | -7.5 |
| Fresh Groundwater | G16 | Sep.2009 | 3 | 110 | | 1214 | 7.9 | 19.9 | | | 214.4 | | 66.1 | 100.1 | 76.3 | 97.8 | 2.8 | 19.5 | 2.68 | -58 | -7.7 |
| Fresh Groundwater | G29 | Sep.2009 | 6 | 60 | | 1291 | 8.1 | 20.9 | | | 255.6 | 26.3 | 57.5 | 69.3 | 8.0 | 205.9 | 12.3 | 14.6 | 0.20 | -51 | -6.6 |
| Brackish Groundwater | G29 | Aug.2010 | 6 | 60 | 6.2 | 3220 | 7.2 | 24.1 | 12 | 5.6 | 803.4 | 143.0 | 264.7 | 178.6 | 386.8 | 189.3 | 4.3 | 77.6 | 6.95 | -50 | -6.5 |
| Brackish Groundwater | G16 | Aug.2010 | 3 | 110 | | 1733 | 7.3 | 21.5 | 16 | 4.8 | 766.8 | 5.5 | 67.1 | 205.4 | 246.2 | 197.8 | 4.5 | 37.9 | 4.00 | -56 | -7.8 |
| Brackish Groundwater | G9 | Jun.2008 | 9 | 104 | | 3110 | 7.8 | 15.0 | | | 823.6 | 47.4 | 110.5 | 85.1 | 255.1 | 222.2 | 5.1 | 36.7 | 7.40 | -47 | -5.3 |
| Brackish Groundwater | G9 | Sep.2009 | 9 | 104 | | 3190 | 8.5 | 20.4 | | | 917.4 | 65.8 | 116.6 | 61.6 | 279.8 | 245.1 | 5.7 | 40.6 | 8.02 | -51 | -6.2 |
| Brackish Groundwater | G9 | Aug.2010 | 9 | 104 | | 4600 | 6.3 | 24.3 | 18 | 4.0 | 1228.3 | 337.4 | 296.3 | 92.3 | 392.2 | 455.4 | 5.5 | 45.3 | 11.59 | -48 | -6.6 |
| Brackish Groundwater | G14' | Aug.2010 | 5 | 110 | 2.1 | 2850 | 6.1 | 22.9 | 120 | 1.8 | 553.8 | 433.7 | 491.4 | 134.0 | 186.0 | 312.8 | 52.3 | 96.6 | 1.06 | -53 | -6.8 |
| Brackish Groundwater | G13 | Aug.2010 | 3 | 90 | | 3230 | 8.9 | 19.2 | 36 | 2.4 | 908.8 | 210.6 | 231.8 | 92.3 | 92.0 | 413.5 | 29.0 | 115.7 | 0.26 | -45 | -5.2 |
| Brackish Groundwater | G13 | Jun.2008 | 3 | 90 | | 3180 | 7.9 | 13.5 | | | 945.4 | | 122.2 | 52.2 | 51.4 | 351.9 | 25.0 | 105.4 | 0.55 | -45 | -5.1 |
| Brackish Groundwater | G13 | Sep.2009 | 3 | 90 | 2.6 | 3070 | 8.1 | 15.3 | | | 882.0 | | 91.9 | 38.5 | 36.5 | 362.9 | 26.0 | 105.3 | 0.35 | -43 | -5.2 |
| Brackish Groundwater | G2 | Jun.2008 | 2 | 60 | | 3780 | 7.7 | 18.5 | | | 1093.4 | | 200.3 | 96.1 | 67.3 | 484.1 | 25.8 | 103.1 | 1.03 | -46 | -5.1 |
| ***River water samples:*** | | | | | | | | | | | | | | | | | | | | | |
| Fresh water samples | | | | | | | | | | | | | | | | | | | | | |
| Dai River | S9 | Aug.2010 | | | | 511 | 7.2 | 22.2 | 22 | 5.5 | 80.3 | 65.2 | 107.9 | 86.3 | 72.1 | 29.9 | 3.7 | 16.7 | 0.34 | -66 | -9.4 |
| Dai River | S12 | Aug.2010 | | | | 485 | 7.5 | 25.8 | 18 | 7.3 | 71.3 | 41.9 | 83.7 | 83.4 | 54.3 | 27.6 | 4.0 | 15.1 | 0.30 | -69 | -9.7 |
| Dai River | S8 | Aug.2010 | | | | 495 | 7.3 | 22.1 | 24 | 5.8 | 68.6 | 54.9 | 96.8 | 89.3 | 62.3 | 27.2 | 4.1 | 15.9 | 0.32 | -67 | -9.6 |
| Yang River | S6 | Aug.2010 | | | | 507 | 7.0 | 23.3 | 17 | 4.5 | 66.0 | 51.9 | 80.5 | 107.2 | 61.6 | 32.2 | 3.8 | 16.8 | 0.30 | -65 | -9.2 |
| Yang River | S2 | Aug.2010 | | | | 435 | 7.3 | 7.3 | 6 | 4.9 | 63.2 | 40.2 | 92.2 | 101.2 | 59.3 | 29.9 | 4.3 | 14.0 | 0.26 | -71 | -10.1 |
| Yang River | S5 | Sep.2009 | | | | 718 | 8.4 | 24.4 | | | 85.2 | 12.9 | 62.0 | 107.8 | 46.1 | 52.0 | 6.4 | 17.4 | 0.36 | -45 | -5.8 |
| Yang River | S4 | Sep.2009 | | | | 2630 | 8.1 | 24.7 | | | 733.1 | 6.6 | 142.2 | 115.5 | | | | | | -42 | -5.3 |
| Yang River | S6 | Sep.2009 | | | | 718 | 8.3 | 25.6 | | | 99.4 | 6.6 | 57.5 | 107.8 | 44.3 | 59.8 | 5.0 | 16.4 | 0.30 | -43 | -7.2 |
| Dai River | S9 | Sep.2009 | | | 2.0 | 560 | 8.0 | 23.6 | | | 88.8 | 6.8 | 60.5 | 92.4 | 57.2 | 33.2 | 3.3 | 16.7 | 0.36 | -46 | -5.8 |
| Dai River | S8 | Sep.2009 | | | | 1013 | 8.2 | 23.6 | | | 174.0 | 6.4 | 82.2 | 92.4 | 52.1 | 98.1 | 6.0 | 23.6 | 0.55 | -43 | -5.4 |
| Yang River | S6 | Jun.2008 | | | | 1166 | 7.5 | 12.6 | | | 81.7 | 12.6 | 71.3 | 112.5 | 53.6 | 47.5 | 4.9 | 18.0 | 0.31 | -49 | -5.5 |
| Dai River | S10 | Jun.2008 | | | | 1255 | 9.1 | 28.0 | | | 208.9 | 23.1 | 73.7 | 175.7 | 60.7 | 123.2 | 10.8 | 24.2 | 0.57 | -44 | -3.3 |
| Dai River | S9 | Jun.2008 | | | | 1163 | 7.8 | 11.8 | | | 92.3 | 7.5 | 52.2 | 90.6 | 54.7 | 31.1 | 3.6 | 19.5 | 0.35 | -40 | -3.9 |
| Brackish and salt water samples | | | | | | | | | | | | | | | | | | | | | |
| Yang River | S3 | Sep.2009 | | | | 34800 | 7.8 | 15.5 | | | 11289.4 | | 1684.7 | 115.5 | 251.0 | 5658.7 | 231.3 | 690.1 | 4.29 | -21 | -2.4 |
| Dai River | S12 | Sep.2009 | | | | 47100 | 7.5 | 23.7 | | | 16766.3 | | 2416.5 | 77.0 | 412.1 | 10074.3 | 398.2 | 1306.0 | 7.64 | -12 | -1.2 |
| Dai River | S11 | Sep.2009 | | | | 52500 | 8.5 | 23.4 | | | 1601.5 | 2.8 | 258.7 | 84.7 | 79.7 | 801.4 | 32.9 | 102.7 | 0.93 | -41 | -5.2 |
| Yang River | S1 | Jun.2008 | | | | 39800 | 8.7 | 24.2 | | | 14953.5 | | 2035.9 | 134.5 | 313.0 | 7496.0 | 270.5 | 928.3 | 5.55 | -15 | -1.1 |
| Yang River | S2 | Jun.2008 | | | | 20200 | 8.8 | 28.5 | | | 8328.3 | | 912.1 | 189.4 | 233.4 | 4094.2 | 147.9 | 495.9 | 3.37 | -28 | -2.5 |
| Dai River | S7 | Jun.2008 | | | | 49500 | 8.4 | 11.5 | | | 16677.1 | | 2261.3 | 113.9 | 349.0 | 8730.0 | 326.4 | 1084.0 | 6.36 | -11 | -0.6 |
| ***Seawater:*** | SW1 | Aug.2010 | | | | 45600 | 7.8 | 25.5 | 83 | 4.90 | 14768.3 | 810.1 | 4047.0 | 148.8 | 312.7 | 8326.4 | 293.6 | 1007.0 | 5.79 | 3.8 | 1.1 |
| ***Seawater:*** | SW1 | Sep.2009 | | | | 47700 | 7.8 | 24.2 | | | 16568.0 | | 2394.3 | 92.4 | 352.2 | 8922.0 | 322.2 | 1107.0 | 6.45 | -10 | -1.1 |
| | SW2 | Sep.2009 | | | | 39500 | 7.3 | 23.2 | | | 14484.8 | | 1926.6 | 107.8 | 313.2 | 7214.0 | 267.9 | 916.9 | 5.43 | -17 | -2.0 |

**Appendix** for the paper titled "Delineating multiple salinization processes in a coastal plain aquifer, northern China: hydrochemical and isotopic evidence" by Han and Currell

[Figure]

Fig. S1 Maps showing the distribution of groundwater level contours in shallow aquifer (a) in 1986 (from Han, 1988), (b) in 1998 (from Zuo, 2006), (c) in 2004 (from Zuo, 2006), and (d) in 2010 (this study). The depression area refers to the area enclosed by 0 m.a.s.l. contour line of groundwater levels.

[Figure]

Fig. S2 Graph showing the temporal variation of monthly cumulative rainfall departure (CRD, Weber and Stewart, 2004), monthly precipitation, average concentration of chloride in groundwater (dark blue) and surface area with >250 mg Cl/L (yellow) between 1963 and 2008 (data from Zang et al., 2010).

[Figure]

Fig. S3 Graph showing δ²H vs. δ¹⁸O of water samples in rainfall and river water. Dry season- July to October; wet season- November to June.

Table S1. NO$_3$/Cl and Sr/Cl ratios in water samples

| WaterType | ID | Sampling Time | Cl mg/L | NO$_3$ mg/L | Sr mg/L | NO$_3$/Cl | Sr/Cl (×10$^{-3}$) |
|---|---|---|---|---|---|---|---|
| **_Shallow groundwater samples:_** | | | | | | | |
| Fresh Groundwater | G4 | Aug.2010 | 124.3 | 92.6 | 0.67 | 0.745 | 5.41 |
| | G27 | Aug.2010 | 177.5 | 163.0 | 1.22 | 0.918 | 6.88 |
| | G12 | Aug.2010 | 276.9 | 31.9 | 0.25 | 0.115 | 0.90 |
| | G17 | Aug.2010 | 88.8 | 39.5 | 0.33 | 0.445 | 3.68 |
| | G18 | Aug.2010 | 124.3 | 137.8 | 0.71 | 1.109 | 5.71 |
| | G1 | Sep.2009 | 220.1 | 2.0 | 0.66 | 0.009 | 3.00 |
| | G4 | Sep.2009 | 124.3 | 178.5 | 0.85 | 1.437 | 6.83 |
| | G5 | Sep.2009 | 303.4 | 162.3 | 1.14 | 0.535 | 3.74 |
| | G23 | Sep.2009 | 88.8 | 163.4 | 0.55 | 1.841 | 6.17 |
| | G7 | Sep.2009 | 81.7 | 41.2 | 0.41 | 0.505 | 5.01 |
| | G8 | Sep.2009 | 334.6 | 85.4 | 1.19 | 0.255 | 3.57 |
| | G11 | Sep.2009 | 222.9 | 74.5 | 0.84 | 0.334 | 3.75 |
| | G12 | Sep.2009 | 174.0 | 46.0 | 0.53 | 0.264 | 3.06 |
| | G28 | Sep.2009 | 127.8 | 22.5 | 0.58 | 0.176 | 4.54 |
| | G18 | Sep.2009 | 110.1 | 152.9 | 0.65 | 1.389 | 5.94 |
| | G20 | Sep.2009 | 246.9 | 107.1 | 3.94 | 0.434 | 15.94 |
| | G17 | Sep.2009 | 243.5 | 126.4 | 0.71 | 0.519 | 2.91 |
| | G3 | Jun.2008 | 315.0 | | 0.46 | | 1.47 |
| | G4 | Jun.2008 | 74.6 | 75.8 | 0.47 | 1.017 | 6.29 |
| | G5 | Jun.2008 | 310.9 | 119.5 | 1.43 | 0.384 | 4.61 |
| | G8 | Jun.2008 | 349.7 | 55.9 | 1.08 | 0.160 | 3.07 |
| | G12 | Jun.2008 | 281.0 | 29.8 | 0.19 | 0.106 | 0.68 |
| | G17 | Jun.2008 | 227.9 | 92.6 | 0.57 | 0.406 | 2.50 |
| | G18 | Jun.2008 | 115.5 | 144.2 | 0.41 | 1.249 | 3.55 |
| | G20 | Jun.2008 | 234.3 | 18.2 | 2.10 | 0.078 | 8.95 |
| Brackish Groundwater | G1 | Aug.2010 | 312.4 | 120.0 | 0.54 | 0.384 | 1.71 |
| | G3 | Aug.2010 | 654.3 | 106.0 | 0.97 | 0.162 | 1.49 |
| | G15 | Aug.2010 | 435.7 | 181.3 | 1.60 | 0.416 | 3.67 |
| | G26 | Aug.2010 | 784.6 | 339.4 | 1.20 | 0.433 | 1.53 |
| | G11 | Aug.2010 | 646.1 | 414.1 | 1.51 | 0.641 | 2.34 |
| | G20 | Aug.2010 | 596.4 | 177.9 | 3.95 | 0.298 | 6.62 |
| | G5 | Aug.2010 | 447.3 | 253.3 | 0.78 | 0.566 | 1.74 |
| | G8 | Aug.2010 | 596.4 | 141.4 | 1.87 | 0.237 | 3.13 |
| | G7 | Aug.2010 | 299.6 | 338.3 | 0.66 | 1.129 | 2.20 |
| | G19 | Aug.2010 | 600.0 | 211.9 | 7.10 | 0.353 | 11.84 |
| | G24 | Aug.2010 | 646.1 | 952.1 | 2.44 | 1.474 | 3.78 |
| | G22 | Sep.2009 | 408.3 | 2.0 | 0.70 | 0.005 | 1.72 |
| | G10 | Sep.2009 | 2563.1 | 27.7 | 1.92 | 0.011 | 0.75 |
| | G14 | Sep.2009 | 291.1 | 109.8 | 0.82 | 0.377 | 2.81 |
| | G24 | Sep.2009 | 454.4 | 441.5 | 0.93 | 0.972 | 2.05 |
| | G19 | Sep.2009 | 622.0 | 91.5 | 4.87 | 0.147 | 7.83 |
| | G1 | Jun.2008 | 717.1 | 87.7 | 0.17 | 0.122 | 0.24 |
| | G11 | Jun.2008 | 451.2 | | 1.00 | | 2.21 |
| | G14 | Jun.2008 | 372.8 | 267.3 | 0.93 | 0.717 | 2.48 |
| | G15 | Jun.2008 | 1675.6 | 146.8 | 2.17 | 0.088 | 1.29 |

**Deep Groundwater samples:**

| | | | | | | | |
|---|---|---|---|---|---|---|---|
| Fresh Groundwater | G25 | Aug.2010 | 68.1 | 71.0 | 0.25 | 1.042 | 3.66 |
| | G16 | Sep.2009 | 214.4 | | 2.68 | | 12.51 |
| | G29 | Sep.2009 | 255.6 | 26.3 | 0.20 | 0.103 | 0.77 |
| Brackish Groundwater | G29 | Aug.2010 | 803.4 | 143.0 | 6.95 | 0.178 | 8.65 |
| | G16 | Aug.2010 | 766.8 | 5.5 | 4.00 | 0.007 | 5.21 |
| | G9 | Jun.2008 | 823.6 | 47.4 | 7.40 | 0.058 | 8.98 |
| | G9 | Sep.2009 | 917.4 | 65.8 | 8.02 | 0.072 | 8.74 |
| | G9 | Aug.2010 | 1228.3 | 337.4 | 11.59 | 0.275 | 9.44 |
| | G14' | Aug.2010 | 553.8 | 433.7 | 1.06 | 0.783 | 1.91 |
| | G13 | Aug.2010 | 908.8 | 210.6 | 0.26 | 0.232 | 0.29 |
| | G13 | Jun.2008 | 945.4 | | 0.55 | | 0.58 |
| | G13 | Sep.2009 | 882.0 | | 0.35 | | 0.39 |
| | G2 | Jun.2008 | 1093.4 | | 1.03 | | 0.94 |

**River water samples:**

Fresh water samples

| | | | | | | | |
|---|---|---|---|---|---|---|---|
| Dai River | S9 | Aug.2010 | 80.3 | 65.2 | 0.34 | 0.813 | 4.20 |
| Dai River | S12 | Aug.2010 | 71.3 | 41.9 | 0.30 | 0.588 | 4.14 |
| Dai River | S8 | Aug.2010 | 68.6 | 54.9 | 0.32 | 0.801 | 4.67 |
| Yang River | S6 | Aug.2010 | 66.0 | 51.9 | 0.30 | 0.786 | 4.53 |
| Yang River | S2 | Aug.2010 | 63.2 | 40.2 | 0.26 | 0.636 | 4.10 |
| Yang River | S5 | Sep.2009 | 85.2 | 12.9 | 0.36 | 0.152 | 4.27 |
| Yang River | S4 | Sep.2009 | 733.1 | 6.6 | | 0.009 | |
| Yang River | S6 | Sep.2009 | 99.4 | 6.6 | 0.30 | 0.067 | 3.04 |
| Dai River | S9 | Sep.2009 | 88.8 | 6.8 | 0.36 | 0.077 | 4.03 |
| Dai River | S8 | Sep.2009 | 174.0 | 6.4 | 0.55 | 0.037 | 3.17 |
| Yang River | S6 | Jun.2008 | 81.7 | 12.6 | 0.31 | 0.154 | 3.82 |
| Dai River | S10 | Jun.2008 | 208.9 | 23.1 | 0.57 | 0.111 | 2.72 |
| Dai River | S9 | Jun.2008 | 92.3 | 7.5 | 0.35 | 0.081 | 3.81 |

Brackish and salt water samples

| | | | | | | | |
|---|---|---|---|---|---|---|---|
| Yang River | S3 | Sep.2009 | 11289.4 | | 4.29 | | 0.38 |
| Dai River | S12 | Sep.2009 | 16766.3 | | 7.64 | | 0.46 |
| Dai River | S11 | Sep.2009 | 1601.5 | 2.8 | 0.93 | 0.002 | 0.58 |
| Yang River | S1 | Jun.2008 | 14953.5 | | 5.55 | | 0.37 |
| Yang River | S2 | Jun.2008 | 8328.3 | | 3.37 | | 0.40 |
| Dai River | S7 | Jun.2008 | 16677.1 | | 6.36 | | 0.38 |
| **Seawater:** | SW1 | Aug.2010 | 14768.3 | 810.1 | 5.79 | 0.055 | 0.39 |
| | SW1 | Sep.2009 | 16568.0 | | 6.45 | | 0.39 |
| | SW2 | Sep.2009 | 14484.8 | | 5.43 | | 0.37 |

Note: $NO_3/Cl$ and $Sr/Cl$ – mass ratios

---

## Author Response (AR2)

**Responses to Editor and Reviewer**

Dr. Dongmei HAN

Key Laboratory of Water Cycle & Related Land Surface Processes

Institute of Geographic Sciences and Natural Resources Research

Chinese Academy of Sciences

Beijing 100101, P.R. China

Fax: +86-10-64889849

Tel: +86-10-64889367

Beijing, 11 May, 2018

To: The Editor
Hydrology and Earth System Sciences

Dear Editor,

We are submitting the revised manuscript titled "Delineating multiple salinization processes in a coastal plain aquifer, northern China: hydrochemical and isotopic evidence" (Manuscript ID: hess-2017-617) to *Hydrology and Earth System Sciences.* We made revisions based on the comments provided by the reviewer. We gratefully acknowledge the editor's and the anonymous reviewer's generous help.

Our detailed response, including changes made, as a result of the constructive suggestions made by the reviewer are detailed below. We have updated a revised version of the manuscript in both 'tracked changes' form, and a 'clean' version with all changes accepted:

**Response to Reviewer #2's comments:**

(1) Abstract: The abstract is a reasonable description of what the paper is about, however it would benefit from some more details. Specifically, there are a lot of qualitative terms in the description of major ion geochemistry - put the key values in the text where possible.

**Answer:** **Agree, changes made (see lines 20 to 25)**. The abstract has been updated to include the key values for important data types (e.g. chloride concentrations, stable isotopic compositions) along with the existing values for strontium concentrations and Sr/Cl ratios (lines 25-26).

(2) Lines 23, 77, and 511: By mineralization do you mean TDS (try to avoid using multiple terms for the same thing as it gets confusing)?

**Answer:** **Agree, changes made.**

The text has been revised and the term 'TDS' is used consistently to describe the total dissolved ion content of the groundwater:

Line 23: "the geothermal water *with high TDS* (up to 10.6 g/L)".

Line 77: "*the high TDS*-geothermal water".

Lines 510-511: "The upward mixing of *high TDS*-geothermal water".

(3) In the figure captions of Figs 2&12, it should simply be 'water table' not 'groundwater table'. Figure S1. 'groundwater levels' in the caption could be changed to 'water tables'
**Answer:** **Agree, changes made.** Figure captions updated as suggested.

(4) Line 191: Line 191: 'water demands...'? - what water demands? This needs a bit of explanation
**Answer:** **Agree, Changes made.** Line 195 inserted "for agriculture and industrial usage (Meng, 2004)"

(5) Line 342: change 'observed' to 'measured'
**Answer:** **Agree, changes made.**

(6) Line 358: change 'originates in' to 'originates from'
**Answer:** **Agree, changes made.**

(7) Figure S1. Hard to read (colourful zone for the depression area would help).
**Answer:** **Agree, changes made.** The depression areas enclosed by the 0 m.a.s.l. water table contour line are now shown in a shaded color version of the figure.

Thank you very much for your time and consideration. If you have any further questions regarding our manuscript, please let us know.

Sincerely Yours,

Corresponding Author:
**Dr. Han Dongmei**
Key Laboratory of Water Cycle & Related Land Surface Processes,
Institute of Geographic Sciences and Natural Resources Research,
Chinese Academy of Sciences,
Beijing, 100101,
P.R.China
Tel: +86-10-64889367
Email:dmeihan@gmail.com